# Roto-translated Local Coordinate Frames For Interacting Dynamical Systems

**Miltiadis Kofinas**
University of Amsterdam
m.kofinas@uva.nl

**Naveen Shankar Nagaraja**
BMW Group
Naveen-Shankar.Nagaraja@bmw.de

**Eftratios Gavves**
University of Amsterdam
egavves@uva.nl

## Abstract

Modelling interactions is critical in learning complex dynamical systems, namely systems of interacting objects with highly non-linear and time-dependent behaviour. A large class of such systems can be formalized as *geometric graphs*, *i.e.*, graphs with nodes positioned in the Euclidean space given an *arbitrarily* chosen global coordinate system, for instance vehicles in a traffic scene. Notwithstanding the arbitrary global coordinate system, the governing dynamics of the respective dynamical systems are invariant to rotations and translations, also known as *Galilean invariance*. As ignoring these invariances leads to worse generalization, in this work we propose local coordinate frames per node-object to induce roto-translation invariance to the geometric graph of the interacting dynamical system. Further, the local coordinate frames allow for a natural definition of anisotropic filtering in graph neural networks. Experiments in traffic scenes, 3D motion capture, and colliding particles demonstrate that the proposed approach comfortably outperforms the recent state-of-the-art.

## 1 Introduction

Modelling interacting dynamical systems –systems of interacting objects with highly non-linear and time-dependent behaviour– with neural networks is attracting a significant amount of interest [27, 3, 19] for its potential to learn long-term behaviours directly from observations. A large class of these systems consists of objects in the physical space, for instance pedestrians in a traffic scene [31, 37] or colliding subatomic particles [5]. These systems can be formalized as *geometric graphs*, in which the nodes describe the physical coordinates of the objects among other features. Kipf et al. [27] introduced Neural Relational Inference (NRI) to learn geometric graph dynamical systems using the variational autoencoding framework [25, 38]. Following [27], dynamic NRI [19] advocated sequential latent variable models to encode time-transient behaviours. Both approaches and the majority of learning algorithms for dynamical systems assume an *arbitrary global coordinate system* to encode time-transient interactions and model complex behaviours. In this work, we posit that taking into account the relative nature of dynamics is key to accurate modelling of interacting dynamical systems.

Represented by geometric graphs, learning algorithms of dynamical systems subscribe themselves to the Newtonian space. The absolute notion of Newtonian space and mechanics, however, determines that there exist infinite inertial frames that connect with each other by a rotation and translation. Each of these inertial frames is equivalent in that they can all serve as global coordinate frames and, thus, an arbitrary choice is made. Notwithstanding this arbitrariness, the dynamics of the system are invariant to the choice of a global coordinate frame up to a rotation and translation, in what is also known as *Galilean invariance*. Put otherwise, geometric graphs of interacting dynamical systems often exhibit symmetries that if left to their own devices lead models to subpar learning.

Inspired by the notion of Galilean invariance, we focus on inducing roto-translation invariance in interacting dynamical systems and their geometric graphs to sustain the effects of underlying

pathological symmetries. Symmetries, invariances and equivariances have attracted an increased interest with learning algorithms in the late years [11, 12, 53, 54, 46, 17]. The reason is that with the increasing complexity of new tasks and data, exploiting the symmetries improves sample efficiency [17, 41] by requiring fewer data points and gradient updates. To date the majority of works on exploiting symmetries or data augmentations are with static data [11, 44, 54]. We argue and show that invariance in representations of dynamic data is just as important, if not more, as it is critical in accounting for the inevitable increased pattern complexity and non-stationarity.

We induce roto-translation invariant representations in graphs by local coordinate frames. Each local coordinate frame is centered at a node-object in the geometric graph and rotated to match its angular position –yaw, pitch, and roll. Since all intermediate operations are performed on the local coordinate frames, the graph neural network is roto-translation invariant and the final transformed output is roto-translation equivariant. We obtain equivariance to global roto-translations by an inverse rotation that transforms the predictions back to the global coordinates.

We make the following three contributions. First, we introduce canonicalized roto-translated local coordinate frames for interacting dynamical systems formalized in geometric graphs. Second, by operating solely on these coordinate frames, we enable roto-translation invariant edge prediction and roto-translation equivariant trajectory forecasting. Third, we present a novel methodology for natural anisotropic continuous filters based on relative linear and angular positions of neighboring objects in the canonicalized local coordinate frames. We continue with a brief introduction of relevant background and then the description of our method. We present related work and finish with experiments and ablation studies on a number of settings, including modelling pedestrians in 2D traffic scenes, 3D particles colliding, and 3D human motion capture systems.

## 2 Background

### 2.1 Interacting dynamical systems and geometric graphs

An interacting dynamical system consists of $i = 1, \ldots, N_t$ objects, whose position $\mathbf{p} = (p_x, p_y, p_z)^\top$ and velocity $\mathbf{u} = \frac{d\mathbf{p}}{dt} = (u_x, u_y, u_z)^\top$ in the Euclidean space are recorded over time $t$. The state of the $i$-th object at timestep $t$ is described by $\mathbf{x}_i^t = [\mathbf{p}_i^t, \mathbf{u}_i^t]$, adopting for clarity a column vector notation and using $[\cdot, \cdot]$ to denote vector concatenation.

Over the past few years, a natural way that has emerged for organizing interacting dynamical systems is by geometric graphs [3, 27, 19] through space and time, $\mathcal{G} = \{\mathcal{G}^t\}_{t=1}^T$, where $\mathcal{G}^t = (\mathcal{V}^t, \mathcal{E}^t)$ is the snapshot of the graph at timestep $t$. The nodes $\mathcal{V}^t = \{v_1^t, \ldots, v_{N_t}^t\}$ of the graph correspond to the objects in the dynamical system, with $v_i^t$ corresponding to the state of the $i$-th object, $\mathbf{x}_i^t$. The edges $\mathcal{E}^t \subseteq \{e_{j,i}^t = (v_j^t, v_i^t) \mid (v_j^t, v_i^t) \in \mathcal{V}^t \times \mathcal{V}^t\}$ of the graph, correspond to the interactions from node-object $j$ to node-object $i$. We use $\mathcal{N}(i)$ to denote the graph neighbours of node $v_i$. In the absence of domain knowledge about how objects connect, for instance the links between atoms in molecules, the graph is fully connected. Explicit inference of the graph structure can be achieved by using latent edges $\mathbf{z}_{j,i}^t$ corresponding to the edges $e_{j,i}^t$.

Graph neural networks [42, 32, 18] exchange messages between neighbors and update the vertex and edge embeddings per layer, commonly referred to as message passing

$$\mathbf{h}_{j,i}^{(l)} = f_e^{(l)}\left(\left[\mathbf{h}_i^{(l-1)}, \mathbf{h}_{j,i}^{(l-1)}, \mathbf{h}_j^{(l-1)}\right]\right) \tag{1}$$

$$\mathbf{h}_i^{(l)} = f_v^{(l)}\left(\mathbf{h}_i^{(l-1)}, \underset{j \in \mathcal{N}(i)}{\square} \mathbf{h}_{j,i}^{(l)}\right), \tag{2}$$

where $\mathbf{h}_i^{(l)}$ is the embedding of node $v_i$ at layer $l$ and $\mathbf{h}_{j,i}^{(l)}$ is the embedding of edge $e_{j,i}$ at layer $l$. $f_e, f_v$ denote differentiable functions such as MLPs and $\square$ denotes a permutation invariant function, commonly a summation or an average. Many graph neural networks rely on isotropic filters [26], although various ways [47, 36] to circumvent this constraint have also been explored.

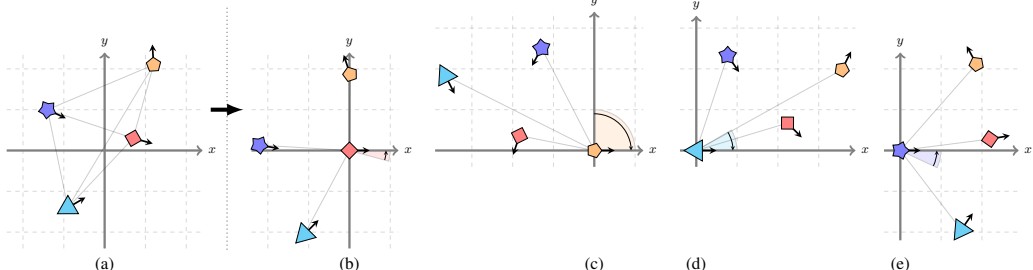

Figure 1: In (a), objects positioned in an arbitrary global 2D coordinate frame; arrows represent orientations. In (b)-(e), objects in the canonicalized local coordinate frames, translated to match the target object's position and rotated to match its orientation

## 2.2 Neural relational inference

Kipf et al. [27] proposed neural relational inference (NRI), a variational autoencoding inference model [25, 38] that explicitly infers the graph structure over a discrete latent graph and simultaneously learns the dynamical system. Given the input trajectories the encoder learns to infer interactions as latent edges $\mathbf{z}_{j,i} \in [0,1]^K$, sampled from a concrete distribution [34, 23] with $K$ edge types. The decoder receives the inferred interactions and the past trajectories, and learns the dynamical system, $p_\theta(\mathbf{x}|\mathbf{z}) = \prod_{t=1}^T p_\theta(\mathbf{x}^{t+1}|\mathbf{x}^{1:t}, \mathbf{z})$. The encoder is a regular graph neural network without explicitly taking time into account and learns to infer latent interactions by maximizing the evidence lower bound for the next time step. The decoder is another graph neural network that either assumes Markovian dynamics $p_\theta(\mathbf{x}^{t+1}|\mathbf{x}^{1:t}, \mathbf{z}) = p_\theta(\mathbf{x}^{t+1}|\mathbf{x}^t, \mathbf{z})$ or is recurrent through time. As the prior and encoder in NRI assume static interactions (*e.g.* whether forces between charged particles are attractive or repulsive), dynamic NRI [19] replaces them with a sequential relation prior based on past states, $p_\phi(\mathbf{z}|\mathbf{x}) = \prod_{t=1}^T p_\phi(\mathbf{z}^t|\mathbf{x}^{1:t}, \mathbf{z}^{1:t-1})$, and an approximate relation posterior based on both the past and future states. The decoder is also reformulated as $p_\theta(\mathbf{x}|\mathbf{z}) = \prod_{t=1}^T p_\theta(\mathbf{x}^{t+1}|\mathbf{x}^{1:t}, \mathbf{z}^{1:t})$, taking into account the dynamic nature of interactions.

## 2.3 Invariance and equivariance

Last, we give a very brief introduction to invariance and equivariance. A function $f : \mathcal{X} \to \mathcal{Y}$ is equivariant [54] under a group of transformations if every transformation $\pi \in \Pi$ of the input $\mathbf{x} \in \mathcal{X}$ can be associated with a transformation $\psi \in \Psi$ of the output, $\psi[f(\mathbf{x})] = f(\pi[\mathbf{x}])$. A special case is invariance, where $\Psi = \{\mathbb{I}\}$, the identity transformation, $f(\mathbf{x}) = f(\pi[\mathbf{x}])$.

In this work, we are interested in translation, *i.e.*, $f(\mathbf{x}) + \boldsymbol{\tau} = f(\mathbf{x} + \boldsymbol{\tau})$ with the translation vector $\boldsymbol{\tau}$, and rotation invariance/equivariance, *i.e.*, $\mathbf{Q}f(\mathbf{x}) = f(\mathbf{Q}\mathbf{x})$ using the rotation matrix $\mathbf{Q}$.

# 3 Roto-translation invariance with local coordinate frames

In this section we present our method, termed LoCS (**Lo**cal **C**oordinate frame**S**). We start with the derivation of roto-translated local coordinate frames and continue with the formulation of graph networks and continuous anisotropic filters operating in these frames.

## 3.1 Local coordinate frames

Starting from the spatio-temporal graph $\mathcal{G}$, we focus for clarity on pairs of node-objects in the same time step, $\mathbf{x}_i^t, \mathbf{x}_j^t$. In the real world, objects are not point particles and have a spatial extension. Central to our method is the use of the angular positions $\boldsymbol{\omega} = (\theta, \phi, \psi)^\top$, otherwise known as yaw, pitch and roll, that describe the orientation of a rigid body with respect to the axes of the coordinate system. We, thus, augment the states $\mathbf{x}_i^t$ with the angular positions, using $\mathbf{v}_i^t = [\mathbf{p}_i^t, \boldsymbol{\omega}_i^t, \mathbf{u}_i^t]$ to denote the augmented state that captures the angular position as well as the linear position and velocity.

Our method draws inspiration from Galilean invariance and inertial frames of reference that capture the relative locations and motions of all the objects in a system. We introduce $N_t$ local coordinate frames, one per object in the system. For the $i$-th object, a local coordinate frame is one in which both the linear and the angular position lie on the origin. By the adoption of the local coordinates with respect to object-centric frames, the behaviors between objects will not depend on the arbitrary positions of objects in the absolute Newtonian space. In other words, the local coordinates offer invariance to global translations and rotations and do not bias the learning algorithm. Our goal, thereby, is to compute the relative local coordinates of all objects $j = 1, \ldots, N_t$, while iterating over the $i$-th reference object.

The transformation from global to local coordinate systems is in two steps. Per target node, we first translate the origin to match its linear position by a translation transformation. Since velocities and angular positions are translation invariant, we only need to perform the translation to the linear positions. This gives us the relative positions $\mathbf{r}_{j,i}^t = [\mathbf{p}_j^t - \mathbf{p}_i^t]$. Then, we canonicalize the local coordinate frame to match the target object's orientation by a rotation transformation, described by the rotation matrix $\mathbf{Q}(\boldsymbol{\omega}_i)$. The coordinates of all other objects are analogously transformed given the $i$-th local coordinate frame. The rotations are performed independently and equivalently to the state components of the $j$-th object, namely the relative –due to the translation transformation performed first– linear positions, the angular positions and the velocities. A schematic overview of the proposed transformation in a 2D setting is presented in fig. 1. Using tensor operations, we compactly write down the transformed state as:

$$\tilde{\mathbf{R}}(\boldsymbol{\omega}) = \mathbf{Q}(\boldsymbol{\omega}) \oplus \mathbf{Q}(\boldsymbol{\omega}) \oplus \mathbf{Q}(\boldsymbol{\omega}) \tag{3}$$

$$\mathbf{v}_{j|i}^t = \tilde{\mathbf{R}}_i^{t\top} [\mathbf{r}_{j,i}^t, \boldsymbol{\omega}_j^t, \mathbf{u}_j^t] \tag{4}$$

where $\oplus$ denotes the direct sum operator that concatenates two matrices along the diagonal resulting in a block diagonal matrix and $\tilde{\mathbf{R}}_i^{t\top} = \tilde{\mathbf{R}}^\top(\boldsymbol{\omega}_i^t)$ to reduce notation clutter. The rotation matrix in eq. (3) has three entries. Each $\mathbf{Q}(\boldsymbol{\omega})$ independently transforms the relative linear positions, the angular positions and the velocities.

**Local coordinate frames in 2D**  We start with the simpler case where the dynamical system resides in the 2D Euclidean space, for instance pedestrians in a traffic scene. In 2 dimensions, the angular position is a scalar value, namely the yaw angle $\theta$. Thus, the rotation matrix for a target node $v_i$ is:

$$\mathbf{Q}(\boldsymbol{\omega}_i) = \mathbf{Q}(\theta_i) = \begin{pmatrix} \cos\theta_i & -\sin\theta_i \\ \sin\theta_i & \cos\theta_i \end{pmatrix} \tag{5}$$

**Local coordinate frames in 3D**  When in the 3D Newtonian space the chain of transformations is the same; first, we translate the origin to match each target node's linear position, and then we rotate the local coordinate frames to match each object's orientation. The translation transformation is identical to the 2D case. However, the rotation transformation is more involved. For the 3D case, we must decompose $\mathbf{Q}(\boldsymbol{\omega})$ into 3 chained elemental rotations, described by the matrices $\mathbf{Q}_z(\theta)$, $\mathbf{Q}_y(\phi)$ and $\mathbf{Q}_x(\psi)$. $\mathbf{Q}_z(\theta)$ describes a rotation around the $z$-axis by an angle $\theta$, $\mathbf{Q}_y(\phi)$ describes a rotation around the $y$-axis by an angle $\phi$ and $\mathbf{Q}_x(\psi)$ describes a rotation around the $x$-axis by an angle $\psi$. We use the following convention that dictates the order of rotations, $\mathbf{Q}(\boldsymbol{\omega}) = \mathbf{Q}_z(\theta)\mathbf{Q}_y(\phi)\mathbf{Q}_x(\psi)$. Each elemental rotation matrix has similar structure to eq. (5). We provide the complete description of all rotation matrices in appendix A.2. After the computation of the rotation matrix, the states are transformed identically to eq. (4).

The local coordinate frames are invariant with respect to global translations and rotations, either in 2 or 3 dimensions. We provide the detailed derivations of $\mathbf{Q}(\boldsymbol{\omega})$ for the 2D and 3D case in appendices A.1 and A.2, respectively, and a detailed proof in appendix A.3.

## 3.2  Approximate angular positions

In practice, we do not always have perfect information about the object states, such as the angular positions. In this case, we can approximate them using the angles of the velocity vector as a proxy. Specifically, in 2 dimensions, the angular position is a scalar value and is approximated using the azimuth angle of the polar representation of the velocity vector, $\theta = \tan^{-1}(u_y/u_x)$. In 3 dimensions, we transform velocities to spherical coordinates $(u_\rho, u_\theta, u_\phi)$ and use these angles to rotate the local

coordinate frame and approximate 2 out of the 3 angles. The angle $\theta$ is computed as above and $\phi = \cos^{-1}(u_z/\|\mathbf{u}\|_2)$. In this case, we retain invariance for the coordinates for which we have perfect knowledge, while we will have an invariance leakage for the approximate ones. That said, experiments show there is little to no consequences and accurate predictions are still attained.

### 3.3 Local coordinate frame graph neural networks

Having canonicalized the object states in the interacting system, we obtain representations that are invariant to global translations and rotations. Following [19], we formulate the core of the network as a variational autoencoder [25, 38] with latent edge types that change dynamically over time. The network receives the canonicalized representations as input and operates solely on the local coordinate systems. We infer the graph structure over a discrete latent graph and simultaneously learn the dynamical system. Learning the graph structure is a roto-translation invariant task; we want to predict the same edge distribution for each pair of vertices regardless of the global rotation of translation. In contrast, trajectory forecasting is a roto-translation equivariant task; a global translation and rotation to the input trajectories should affect the output trajectories equivalently. Following [19], we maximize the evidence lower bound, $\mathcal{L}(\phi, \theta) = \mathbb{E}_{q_\phi(\mathbf{z}|\mathbf{x})}[\log p_\theta(\mathbf{x}|\mathbf{z})] - \mathrm{KL}[q_\phi(\mathbf{z}|\mathbf{x})||p_\phi(\mathbf{z}|\mathbf{x})]$. We provide the exact form of the loss components in appendix B.1.

**Encoder and prior** Our encoder and prior closely follow [19], described in section 2.2. First, we compute the local coordinate frame representations $\mathbf{v}_{j|i}^t$ per pair $j, i$ (including self-loops) and per timestep $t$ according to eq. (4). We then perform a number of message passing steps and obtain a feature vector per object pair. Omitting time indices for clarity, we have

$$\mathbf{h}_{j,i}^{(1)} = f_e^{(1)}\left(\left[\mathbf{v}_{j|i}, \mathbf{v}_{i|i}\right]\right) \tag{6}$$

$$\mathbf{h}_i^{(1)} = f_v^{(1)}\left(g_v^{(1)}\left(\mathbf{v}_{i|i}\right) + \frac{1}{|\mathcal{N}(i)|} \sum_{j \in \mathcal{N}(i)} \mathbf{h}_{j,i}^{(1)}\right) \tag{7}$$

$$\mathbf{h}_{j,i}^{(2)} = f_e^{(2)}\left(\left[\mathbf{h}_i^{(1)}, \mathbf{h}_{j,i}^{(1)}, \mathbf{h}_j^{(1)}\right]\right) \tag{8}$$

The functions $f_e^{(1)}, f_v^{(1)}, f_e^{(2)}$ are MLPs and $g_v^{(1)}$ is a linear layer. We feed the embeddings $\mathbf{h}_{j,i}^{(2)}$ into 2 LSTMs [22]: one forward in time that computes the prior and one backwards in time for the encoder. The hidden state from the forward LSTM is used to compute the prior distribution, while the hidden states from both the forward and the backward LSTM are concatenated to compute the encoder distribution. The formulation is identical to [19]; the exact details can be found in appendix B.1.

During training, we sample interactions $\mathbf{z}_{j,i}^t$ from $q_\phi\left(\mathbf{z}_{j,i}^t|\mathbf{x}\right)$ using Gumbel-Softmax [34, 23]. We perform teacher-forcing during the whole training, and task the model to predict the trajectories only for one step ahead. During inference, we sample interactions from the prior distribution.

**Decoder** As mentioned in section 2.2, the decoder $p_\theta(\mathbf{x}|\mathbf{z}) = \prod_{t=1}^T p_\theta(\mathbf{x}^{t+1}|\mathbf{x}^{1:t}, \mathbf{z}^{1:t})$ is tasked with predicting future trajectories given past and present trajectories as well as the predicted relations. As proposed by [19, 27] we can have either a Markovian or a recurrent decoder, depending on the governing dynamics. In many settings, like colliding elementary particles in physics, the governing dynamics satisfy the Markov property, $p_\theta(\mathbf{x}^{t+1}|\mathbf{x}^{1:t}, \mathbf{z}^{1:t}) = p_\theta(\mathbf{x}^{t+1}|\mathbf{x}^t, \mathbf{z}^t)$. In this case, the decoder is implemented with a graph neural network similar to [19]. In many real-world applications, however, the Markovian assumption does not hold. In that case, the graph neural network also features a GRU unit that learns a recurrent hidden state during the message passing.

We can use local coordinates frames with both types of decoders, as defined in [19, 27], with the difference that the message passing is performed with the local coordinate frame representations $\mathbf{v}_{j|i}^t$, so that we attain roto-translation invariance,

$$\mathbf{m}_{j,i}^t = \sum_k z_{(j,i),k}^t f^k\left(\left[\mathbf{v}_{j|i}^t, \mathbf{v}_{i|i}^t\right]\right) \tag{9}$$

$$\mathbf{m}_i^t = f_v^{(3)}\left(g_v^{(3)}\left(\mathbf{v}_{i|i}^t\right) + \frac{1}{|\mathcal{N}(i)|} \sum_{j \in \mathcal{N}(i)} \mathbf{m}_{j,i}^t\right). \tag{10}$$

The output of the model per time step comprises position and velocity predictions for the next time step. As common in the literature [19], we predict the difference in position and velocity from the previous time step, which equals the velocity and acceleration respectively, and numerically integrate to make predictions. While computations up to the output layer are roto-translation invariant, the predictions must be roto-translation equivariant, so that global roto-translations to the inputs affect the outputs equivalently. To transform predictions back to the global coordinate frame and achieve roto-translation equivariance, we do an inverse rotation by $\mathbf{R}(\boldsymbol{\omega}_i^t) = \mathbf{Q}(\boldsymbol{\omega}_i^t) \oplus \mathbf{Q}(\boldsymbol{\omega}_i^t)$, and then integrate numerically, $i.e.$, $\mathbf{x}_i^{t+1} = \mathbf{x}_i^t + \mathbf{R}(\boldsymbol{\omega}_i^t) \cdot \boldsymbol{\Delta}\mathbf{x}_i^{t+1}$. We provide the definitions for both decoders in appendix B.1, and a detailed proof on equivariance to global roto-translations in appendix A.3.

### 3.4 Anisotropic filtering

One of the main reasons why filters in graph neural networks are isotropic is the inherent absence of an invariant coordinate frame. In a geometric graph dynamical system, object positions can serve this role. We, thus, use the local roto-translation invariant coordinate frames for anisotropic filtering, using weights that depend on the relative linear positions and angular positions of objects given the central $i$-th object. Similar to [45], our filter generating network, implemented by an MLP, is a matrix field that maps relative positions and orientation tuples to graph network filters, $i.e.$ weight matrices, $\mathbf{W}_{\mathcal{F}} : \mathbb{R}^D \times \mathcal{S}^{|\boldsymbol{\omega}|} \to \mathbb{R}^{D_{\text{out}} \times D_{\text{in}}}$. The anisotropic filters replace the isotropic ones in updating the latent edge representations, weighing neighbors according to their positions,

$$\mathbf{h}_{j,i}^t = \mathbf{W}_{\mathcal{F}}\big(\big[\mathbf{Q}^\top(\boldsymbol{\omega}_i^t) \cdot \mathbf{r}_{j,i}^t, \mathbf{Q}^\top(\boldsymbol{\omega}_i^t) \cdot \boldsymbol{\omega}_j^t\big]\big) \cdot \big[\mathbf{v}_{j|i}^t, \mathbf{v}_{i|i}^t\big].$$

### 3.5 Data normalization

While the graph neural networks are roto-translation invariant and equivariant in the intermediate and the output layers respectively, we must also make sure that the pre- and post-processing of data are appropriate. The common practices of min-max normalization or z-score normalization are unsuitable because they anisotropically scale and translate the input and output position and velocities. That is, these transformations change the directions of the velocity vectors non-equivalently. This is counter-intuitive, since velocities are not treated as geometric entities but as generic additional dimensions to the features. For instance, as translation equals a vector subtraction changing the magnitude and the direction of vectors, translating the velocities removes any notion of speed from the input to the neural network. What is more, the scaling operations apply anisotropic transformations, affecting each axis differently.

We instead opt for a much simpler data normalization scheme that is more geometrically oriented and suitable for roto-translation invariance with local coordinate frames. This scheme does not perform any translation operations, since local coordinate frames naturally tend to center data around the origin; besides, they are invariant to a mere isotropic translation to the node positions. For the scaling operation, we opt for a simple isotropic transformation that shrinks relative positions and velocities equivalently across all axes. We scale the inputs, both positions and velocities, by the maximum speed (velocity norm) in the training set, $s_{\max} = \max_i \|\mathbf{u}_i\|$, that is $\mathbf{x}' = \mathbf{S}^{-1}\mathbf{x}$, where $\mathbf{S} = \text{diag}(s_{\max} \cdot \mathbf{1})$. During post-processing, we can convert our predictions to actual units, $e.g.$ $m$ and $m/s$, by applying the inverse transformation, $\mathbf{x} = \mathbf{S}\mathbf{x}'$. We term this operation *speed normalization*.

## 4 Related work

**Learning dynamical systems & trajectory forecasting**   In the late years, and alongside NRI [27] and dNRI [19], many have studied learning dynamical systems [3, 40]. Further, many works have focused on the problems of pedestrian motion prediction and traffic scene trajectory forecasting [1, 28, 21, 35, 39]. A number of works [1, 39] uses distance-based heuristics to create the graph adjacency and estimate interactions. Kosaraju et al. [28] use self-attention to predict the influence of neighbouring nodes. Both approaches are different from our work, since we explicitly predict the latent graph structure and perform inference on it.

**Equivariant deep learning**   Equivariant neural networks [11, 12, 53, 54, 46] have risen in popularity over the past few years, demonstrating high effectiveness and parameter efficiency. Schütt et al. [44] use radial basis functions on pair-wise node distances to generate continuous filters and perform mes-

sage passing using depth-wise separable convolutions. Fuchs et al. [17] introduce SE(3)-transformers by incorporating spherical harmonics in a transformer network, resulting in a 3D roto-translation equivariant attention network. de Haan et al. [13] propose anisotropic gauge equivariant kernels for graph networks on meshes based on neighbouring vertex angles and parallel transport. Walters et al. [50] propose rotationally equivariant continuous convolutions for 2D trajectory prediction. Closer to our work is the work of Satorras et al. [41]. They propose a graph network that leverages the rotation equivariant relative position and roto-translation invariant euclidean distance between node pairs in a novel message passing scheme that updates node features as well as node coordinate embeddings. Different from our work, they do not capitalize on orientations and velocities of neighbouring objects. Our work leverages these impactful quantities expressed in local coordinate frames to make more reliable predictions.

**Anisotropic filtering**   Graph neural networks [42, 32, 18] have been used extensively for modeling dynamical systems [55, 35, 27, 19, 3]. Several works [35] incorporate isotropic graph filters [26, 20] as part of the graph network. However, these isotropic filters have a global weight sharing scheme, *i.e.* they use a single weight matrix for all neighbours, which amounts to a linear transformation over the aggregated neighbour information. Velickovic et al. [47] use self-attention [2] and [36] use Gaussian Mixture Models (GMMs) based on relative neighbour positions.

Other works address this issue by proposing continuous filters on graphs and point clouds [24, 51, 45, 50, 44, 16]. Highly related to our work is the work of Simonovsky and Komodakis [45]. They generalize convolution to arbitrary graphs and introduce dynamically generated filters based on edge attributes to perform continuous anisotropic convolutions on graph signals and point clouds. Differently, though, they do not operate under roto-translated local coordinate systems. This results in diminished parameter efficiency and weight sharing that they compensate for with data augmentation.

# 5    Experiments

We evaluate the proposed method, LoCS, on 2D and 3D geometric graph dynamical systems from the literature. In 2D, we evaluate on a synthetic physics simulation dataset proposed by dNRI [19] and on traffic trajectory forecasting [4]. In 3D, we evaluate on an 3D-extended version of the charged particles [27] and on a motion capture dataset [10]. We compare with NRI [27], dNRI [19], and the very recent EGNN [41]. For all methods we use publicly available code from [19, 41]. For EGNN, we autoregressively feed the output as the input to the next timestep. The full implementations details are in appendix B.3. Our code, data, and models will be available online[1].

Our architecture closely follows dNRI[26, 19]. Unless otherwise specified, all common layers have the same structure, and we use the same number of latent edge types. Following [26, 19], we report the mean squared error of positions and velocities over time. We compute errors in the original unnormalized data space for a fair comparison across different data normalization techniques and scales, $E(t) = \frac{1}{ND} \sum_{n=1}^{N} \|\mathbf{x}_n^t - \hat{\mathbf{x}}_n^t\|_2^2$. We also report the $L_2$ norm errors for positions (displacement errors), $E_p(t) = \frac{1}{N} \sum_{n=1}^{N} \|\mathbf{p}_n^t - \hat{\mathbf{p}}_n^t\|_2$, and velocities, $E_u(t) = \frac{1}{N} \sum_{n=1}^{N} \|\mathbf{u}_n^t - \hat{\mathbf{u}}_n^t\|_2$, separately to gain insights. $L_2$ norm errors are in the same scale with the respective variables (position and velocity) and, thus, more interpretable than MSE. We always ran experiments using 5 different random initialization seeds and report the mean and the standard deviation. We plot each method with a different color, as well as a different marker for color-blind friendly visualizations. The x-coordinates of the markers are chosen for aesthetic purposes; they differ across settings and they are not meant to convey extra information. Here we report the main results and visualizations and provide much more extensive examples in the appendix.

## 5.1    Synthetic dataset

On the synthetic 2D physics simulation we use the same experimental settings as in [19]. The dataset comprises scenes with three particles. Two of the particles move with a constant, randomly initialized velocity and the third particle is initialized with random velocity but pushed away when close to one of the others. Scenes last for 50 timesteps. For evaluation, we use the first 25 timesteps as input and the models are tasked with predicting the following

---

[1]`https://github.com/mkofinas/locs`

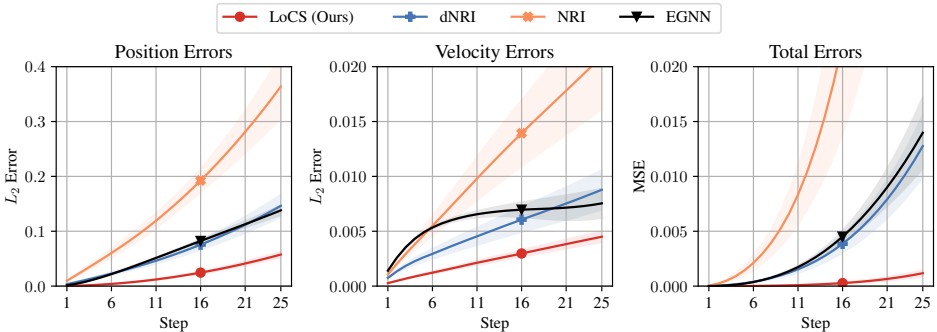

Figure 2: Results on synthetic dataset

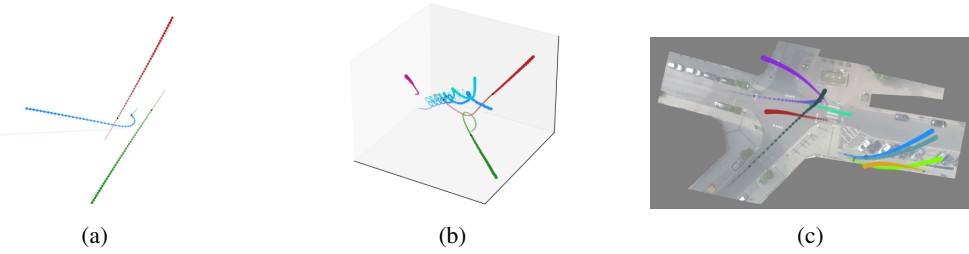

| (a) | (b) | (c) |

Figure 3: LoCS predictions on (a) synthetic dataset, (b) charged particles and (c) inD.

25 timesteps. We report the results in fig. 2 and plot qualitative examples fig. 3a, more in fig. 7 in appendix C.1. We also report the average F1 score for relation prediction in table 1.

In all visualizations, each color denotes a different object. Semi-transparent trajectories indicate the groundtruth future trajectories. Present timesteps are denoted by small black circles. Markers denote the timesteps, increasing in size as trajectories evolve through time. We observe that our method can reliably model interactions and outperforms competing methods in predicting future trajectories.

Table 1: Relation prediction F1 score on synthetic dataset

| Method | NRI | dNRI | LoCS |
|--------|-----|------|------|
| F1 | 26.5 | 60.8 | 88.9 |

## 5.2 Charged particles

In the charged particles datasets the particles interact with one another via electrostatic forces. We extend the dataset by [27] from 2 to 3 dimensions and, following a similar approach to [17, 41], we remove the virtual boxes that confine the particle trajectories. We generate 30,000 scenes for training, 5,000 for validation and 5,000 for testing. Each scene comprises trajectories of 5 particles that carry either a positive or a negative charge. The forces are either attractive or repulsive according to physical laws. Following [27], training and validation scenes last for 49 timesteps. For evaluation, test scenes last for 20 additional timesteps (50 for visualization). We plot the MSE in fig. 4a and $L_2$ errors in Figure 11 in appendix D. LoCS has consistently lower errors, in total as well as individually for positions and velocities. We, further, provide qualitative results for 50 future time steps in fig. 3b, see more in fig. 8 in appendix C.2.

## 5.3 Traffic trajectory forecasting

The inD dataset [4] is a real-world 2D traffic trajectory forecasting dataset of pedestrians, vehicles, and cyclists, recorded at 4 different German traffic intersections. Traffic scenes contain a varying number of participants also changing over time. We follow the exact same setting as in dNRI. The dataset contains 36 recordings; we split them in 19/7/10 for training, validation and testing. We divide each scene into 50-step sequences. We use the first 5 timesteps as input and the model has to predict the remaining 45 time steps. We compare with dNRI, EGNN, and a GRU [7] baseline, but not with NRI since there are varying number of nodes. We plot the MSE in fig. 4b and $L_2$ errors in Figure 12 in appendix D and as before, LoCS outperforms competing methods consistently.

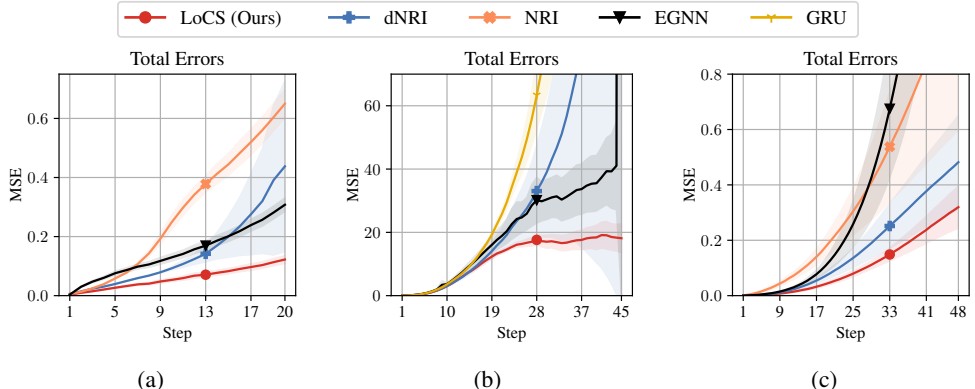

Figure 4: Total error curves in: (a) charged particles, (b) inD, (c) motion #35

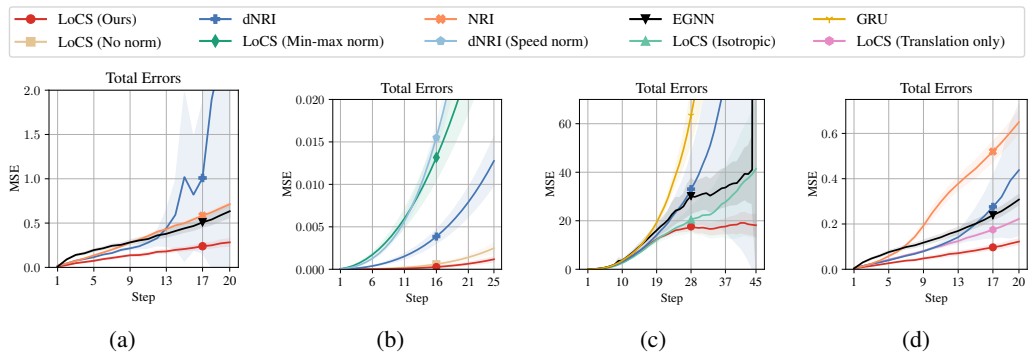

Figure 5: Total error curves in ablation experiments: (a) on highly interactive charged particles, (b) on the impact of speed normalization, (c) on the impact of isotropic filters, (d) on the impact of rotation.

## 5.4 Motion capture

Last, we experiment with the CMU motion capture database [10] with 3D data, following the exact same setting as [27, 19] and studying the motion of subject #35. We train the models using sequences of 50 timesteps as input and evaluate on sequences of 99 time steps. We plot the MSE in fig. 4c and $L_2$ errors in fig. 13 in appendix D and observe that LoCS is attaining consistently lower errors.

## 5.5 Ablation experiments

**High-intensity interactions**  We assess LoCS in fig. 5a in highly interactive scenarios by creating a subset of the charged particles test set (819 scenes – $16.38\%$ of the original test set) in which a simple constant velocity model performs poorly ($L_2$ error $> 1.5$). With more and stronger interactions the proposed local coordinate frames performs even better relatively to competitors.

**Speed normalization impact**  We assess the impact of speed normalization on the synthetic dataset in fig. 5b. We evaluate LoCS with and without speed normalization, as well as dNRI with speed normalization instead of min-max normalization. Results are shown in fig. 5b. We observe that when the inputs are normalized using min-max normalization, LoCS underperforms. Without any normalization, LoCS already performs better than other baselines. Using speed normalization improves the performance of LoCS even further. Finally, speed normalization is not the main cause for the improvements for LoCS. If it were, it should also benefit dNRI. We conclude that speed normalization is important for local coordinate frames to make sure equivariance is maintained after un-normalization. LoCS attains the highest accuracies when combined with speed normalization, although it works quite well even without it, while methods that are not equivariant like dNRI do not benefit from it.

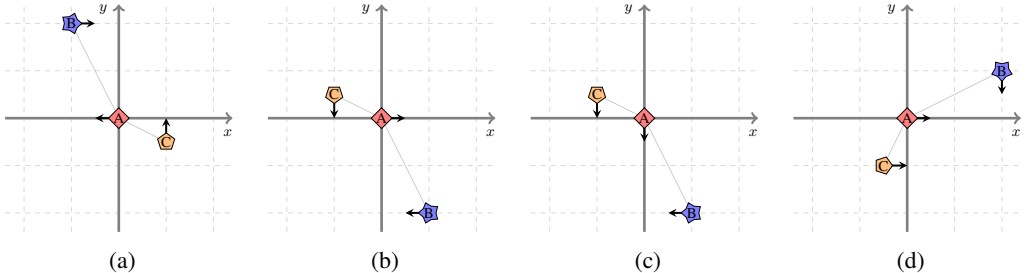

Figure 6: In a, b, c, translated-only local coordinate frames for object A in 3 different dynamical systems #1-#3. In d, dynamical system #3, A's roto-translated local coordinate frame

**Impact of anisotropic filtering**    We compare anisotropic and isotropic filtering on inD in fig. 5c. Be it with isotropic or anisotropic filters, the local coordinate frames outperform the competitors. That said, anisotropic filters give a clear advantage over isotropic ones.

**Impact of rotation in spherical symmetries**    Even though particles do not have intrinsic orientations, they do have the direction of their velocity. We postulate that roto-translated local coordinate frames allow for more efficient learning, as long as the coordinate frames are consistently invariant. To motivate this, consider the simple case of a 2D system shown in fig. 6a with three objects A, B, C with no intrinsic orientation. In fig. 6a, B is far above left of A, while C is near below right of A. Rotating the system by $\pi$, see fig. 6b, A still lies at the same origin, however, C is now in the top left and B is in the bottom right quadrant. Without canonicalizing with respect to rotations, the description of the two systems is very different, and subsequent neural networks will have to learn to account for this underlying symmetry. On the other hand, different dynamical systems should yield different representations. For example, consider the 2 dynamical systems shown in figs. 6b and 6c. While they only differ in A's velocity, their dynamics vary greatly: in fig. 6c, A and B are moving perpendicularly to one another and may crush due to attractive forces. In the canonicalized frame in fig. 6d, the neighbour representations are indeed very different from fig. 6b.

In the end, we care that our inputs are represented consistently (invariantly) if their relative differences (translations or rotations) are the same according to the system at hand, regardless of how we obtain the reference axis for the rotation (intrinsic angular position or another invariant quantity like the angles of acceleration vectors). Thus, applying both translation and rotation transformation helps even for objects with no intrinsic viewpoint and orientation, like point masses. We confirm the hypothesis in an ablation experiment with charged particles, where using only the translation transformation to form local coordinate frames leads to decreased accuracy, see fig. 5d.

## 6   Conclusion

In this work we introduced LoCS, a method that introduces canonicalized roto-translated local coordinate frames for all objects in interacting dynamical systems formalized in geometric graphs. These coordinate frames grant us global invariance to roto-translations and naturally allow for anisotropic continuous filtering. We demonstrate the effectiveness of our method in a range of 2D and 3D settings, outperforming recent state-of-the-art works.

**Limitations**    Many dynamical systems in nature are not formalized as geometric graphs (*e.g.* social networks), or are not intrinsically described by angular positions (*e.g.* elementary particles), in which case the proposed method is not applicable. Furthermore, although we approximate angular positions using velocities, our method guarantees full invariance/equivariance to global roto-translations only in the 2D case, while in 3 dimensions we have an equivariance leakage. We have identified two modes where local coordinate frames do not exhibit the same large improvements. First, when the data setting relies, in fact, on a global coordinate frame, like different positions in a basketball court [56]. Second, when the direction of the local coordinate frames cannot be well-defined, like objects with **0** velocity (although simply and arbitrarily setting the orientation to **0** works just as well). Last, a theoretical limitation is that Euler angles, which LoCS also uses, are prone to singularities and gimbal lock, although in practice we observed no problem.

## Acknowledgments

The project is funded by the NWO LIFT grant 'FLORA'.

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
