# A  Roto-translation invariance

## A.1  Rotations in 2 dimensions

In 2-dimensional settings, there exists a single scalar angular position, the yaw angle $\theta$. Following eqs. (3) and (5), we compute the rotation matrices $\mathbf{Q}$, $\mathbf{R}$ and $\tilde{\mathbf{R}}$ as follows:

$$\mathbf{Q}(\theta) = \begin{pmatrix} \cos\theta & -\sin\theta \\ \sin\theta & \cos\theta \end{pmatrix} \tag{11}$$

$$\mathbf{R}(\theta) = \mathbf{Q}(\theta) \oplus \mathbf{Q}(\theta) = \begin{pmatrix} \mathbf{Q}(\theta) & \mathbf{0}_{2\times2} \\ \mathbf{0}_{2\times2} & \mathbf{Q}(\theta) \end{pmatrix} \tag{12}$$

$$\tilde{\mathbf{R}}(\theta) = \mathbf{Q}(\theta) \oplus \mathbf{Q}(\theta) \oplus \mathbf{Q}(\theta) = \begin{pmatrix} \mathbf{Q}(\theta) & & \mathbf{0} \\ & \mathbf{Q}(\theta) & \\ \mathbf{0} & & \mathbf{Q}(\theta) \end{pmatrix} \tag{13}$$

The second rotation matrix $\mathbf{Q}$ in $\tilde{\mathbf{R}}$ is used to rotate the angular positions. In order to perform the transformation, we have to express the angular positions in a format suitable for linear transformations; we do so by transforming them to rotation matrices, perform a matrix multiplication, and then transform the angular positions back to angle format. In 2 dimensions, we use eq. (11) to convert the angular positions $\theta$ to matrix format. After the rotation, we can convert them back to angle format using the 2-argument arc-tangent function:

$$\theta = \mathrm{atan2}(\sin\theta, \cos\theta) \tag{14}$$

**Simplified rotations**  In 2 dimensions, the computations can be simplified since rotations commute. First, we show that chained rotations result in angle addition/subtraction, that is:

$$\mathbf{Q}(\theta_i) \cdot \mathbf{Q}(\theta_j) = \begin{pmatrix} \cos\theta_i & -\sin\theta_i \\ \sin\theta_i & \cos\theta_i \end{pmatrix} \cdot \begin{pmatrix} \cos\theta_j & -\sin\theta_j \\ \sin\theta_j & \cos\theta_j \end{pmatrix} \tag{15}$$

$$= \begin{pmatrix} \cos\theta_i\cos\theta_j - \sin\theta_i\sin\theta_j & -\cos\theta_i\sin\theta_j - \sin\theta_i\cos\theta_j \\ \sin\theta_i\cos\theta_j + \cos\theta_i\sin\theta_j & -\sin\theta_i\sin\theta_j + \cos\theta_i\cos\theta_j \end{pmatrix} \tag{16}$$

$$= \begin{pmatrix} \cos(\theta_i+\theta_j) & -\sin(\theta_i+\theta_j) \\ \sin(\theta_i+\theta_j) & \cos(\theta_i+\theta_j) \end{pmatrix} \tag{17}$$

$$= \mathbf{Q}(\theta_i + \theta_j) \tag{18}$$

Following the same approach, we compute the inverse rotation:

$$\mathbf{Q}^\top(\theta_i) \cdot \mathbf{Q}(\theta_j) = \mathbf{Q}(-\theta_i) \cdot \mathbf{Q}(\theta_j) = \mathbf{Q}(\theta_j - \theta_i) \tag{19}$$

Thus, instead of rotating the angular positions (expressed in rotation matrix form) using the rotation matrix $\mathbf{Q}$, in practice we perform the transformation directly to the angles via addition/subtraction, and replace the matrix $\mathbf{Q}$ with the identity matrix $\mathbf{I}_{1\times1}$. This results in the following equations that replace eqs. (3) and (4):

$$\tilde{\mathbf{R}}(\theta) = \mathbf{Q}(\theta) \oplus \mathbf{I}_{1\times1} \oplus \mathbf{Q}(\theta) = \begin{pmatrix} \mathbf{Q}(\theta) & & \mathbf{0} \\ & \mathbf{I}_{1\times1} & \\ \mathbf{0} & & \mathbf{Q}(\theta) \end{pmatrix} \tag{20}$$

$$\mathbf{v}_{j|i}^t = \tilde{\mathbf{R}}_i^{t\top}\left[\mathbf{r}_{j,i}^t, \theta_j^t - \theta_i^t, \mathbf{u}_j^t\right] \tag{21}$$

**Angular position approximation**  In order to approximate the yaw angle $\theta$ using the velocity vector $\mathbf{u} = (u_x, u_y)^\top$, we transform the velocities to polar coordinates and use the azimuth angle of the polar representation to compute $\theta$ as follows:

$$\theta = \tan^{-1}\left(\frac{u_y}{u_x}\right) \tag{22}$$

In practice, we use the 2-argument arc-tangent function $\mathrm{atan2}(y, x)$ to compute $\theta$.

Computing the relative angular position can result in angles outside the range $[-\pi, \pi)$, which can lead to discrepancies. Thus, we wrap the computed angle difference so that it always belongs in that range. Furthermore, in all cases that angles are not used geometrically (*e.g.* for rotations), we normalize them by dividing by $\pi$, resulting in an output range of $[-1, 1)$.

## A.2  Rotations 3 dimensions

In 3 dimensions, the computation of rotation matrices is more involved than the 2D case. As described in section 3.1, we decompose the rotation matrix $\mathbf{Q}(\boldsymbol{\omega})$ into 3 chained elemental rotations $\mathbf{Q}_z(\theta)$, $\mathbf{Q}_y(\phi)$ and $\mathbf{Q}_x(\psi)$. The elemental rotation matrices are computed as follows:

$$\mathbf{Q}_z(\theta) = \begin{pmatrix} \cos\theta & -\sin\theta & 0 \\ \sin\theta & \cos\theta & 0 \\ 0 & 0 & 1 \end{pmatrix} \tag{23}$$

$$\mathbf{Q}_y(\phi) = \begin{pmatrix} \cos\phi & 0 & \sin\phi \\ 0 & 1 & 0 \\ -\sin\phi & 0 & \cos\phi \end{pmatrix} \tag{24}$$

$$\mathbf{Q}_x(\psi) = \begin{pmatrix} 1 & 0 & 0 \\ 0 & \cos\psi & -\sin\psi \\ 0 & \sin\psi & \cos\psi \end{pmatrix} \tag{25}$$

Next, we compose the elemental matrices to compute the full rotation matrix:

$$\mathbf{Q}(\boldsymbol{\omega}) = \mathbf{Q}_z(\theta)\mathbf{Q}_y(\phi)\mathbf{Q}_x(\psi) \tag{26}$$

$$= \begin{pmatrix} \cos\phi\cos\theta & \sin\psi\sin\phi\cos\theta - \cos\psi\sin\theta & \cos\psi\sin\phi\cos\theta + \sin\psi\sin\theta \\ \cos\phi\sin\theta & \sin\psi\sin\phi\sin\theta + \cos\psi\cos\theta & \cos\psi\sin\phi\sin\theta - \sin\psi\cos\theta \\ -\sin\phi & \sin\psi\cos\phi & \cos\psi\cos\phi \end{pmatrix} \tag{27}$$

$$\mathbf{Q}^\top(\boldsymbol{\omega}) = \mathbf{Q}_x^\top(\psi)\mathbf{Q}_y^\top(\phi)\mathbf{Q}_z^\top(\theta) \tag{28}$$

$$= \begin{pmatrix} \cos\phi\cos\theta & \cos\phi\sin\theta & -\sin\phi \\ \sin\psi\sin\phi\cos\theta - \cos\psi\sin\theta & \sin\psi\sin\phi\sin\theta + \cos\psi\cos\theta & \sin\psi\cos\phi \\ \cos\psi\sin\phi\cos\theta + \sin\psi\sin\theta & \cos\psi\sin\phi\sin\theta - \sin\psi\cos\theta & \cos\psi\cos\phi \end{pmatrix} \tag{29}$$

$$\mathbf{R}(\boldsymbol{\omega}) = \begin{pmatrix} \mathbf{Q}(\boldsymbol{\omega}) & \mathbf{0}_{3\times3} \\ \mathbf{0}_{3\times3} & \mathbf{Q}(\boldsymbol{\omega}) \end{pmatrix} \tag{30}$$

$$\tilde{\mathbf{R}}(\boldsymbol{\omega}) = \begin{pmatrix} \mathbf{Q}(\boldsymbol{\omega}) & & \mathbf{0} \\ & \mathbf{Q}(\boldsymbol{\omega}) & \\ \mathbf{0} & & \mathbf{Q}(\boldsymbol{\omega}) \end{pmatrix} \tag{31}$$

Similar to the 2D case, in order to rotate the angular positions we have to convert them to a format suitable for linear transformations. We use eqs. (26) and (27) to perform the conversion. After rotation, we convert the angular positions back to angle format. Using $\boldsymbol{\Omega}$ to denote the transformed angular positions expressed in matrix format, we have the following:

$$\boldsymbol{\omega} = \begin{pmatrix} \theta \\ \phi \\ \psi \end{pmatrix} = \begin{pmatrix} \mathrm{atan2}(\boldsymbol{\Omega}_{1,0}, \boldsymbol{\Omega}_{0,0}) \\ \sin^{-1}(-\boldsymbol{\Omega}_{2,0}) \\ \mathrm{atan2}(\boldsymbol{\Omega}_{2,1}, \boldsymbol{\Omega}_{2,2}) \end{pmatrix} \tag{32}$$

We use Pytorch3D [62] for this conversion, specifically the function `matrix_to_euler_angles`, following the ZYX convention.

**Angular position approximation**   Using the velocity angles to approximate angular positions and create the local coordinate frames in 3 dimensions is not as straight-forward as the 2-dimensional case. The spherical coordinates representation of the velocity vector gives us 2 angles instead of the 3 that are required to fully describe a 6-DOF 3D rigid body.

In the following equations, we use the notation convention $(\rho, \theta, \phi)$ to represent the radial distance, azimuthal angle and polar angle, respectively. The transformations from Cartesian to spherical coordinates are as follows:

$$\rho = \sqrt{u_x^2 + u_y^2 + u_z^2} \tag{33}$$

$$\theta = \tan^{-1}\left(\frac{u_y}{u_x}\right) \tag{34}$$

$$\phi = \cos^{-1}\left(\frac{u_z}{\rho}\right) \tag{35}$$

In practice, similar to the 2-dimensional setting, we use the `atan2` function to compute $\theta$. Furthermore, we add $\epsilon = 1e - 8$ to the denominator in eq. (35) and clamp the fraction in the range $[-1, 1]$ to avoid numerical instabilities that may occur, especially during backpropagation.

Having access to 2 angular positions, we compute the rotation matrix $\mathbf{Q}$ as follows:

$$\mathbf{Q}(\boldsymbol{\omega}) \stackrel{\psi=0}{=} \mathbf{Q}_z(\theta)\mathbf{Q}_y(\phi) = \begin{pmatrix} \cos\phi\cos\theta & -\sin\theta & \sin\phi\cos\theta \\ \cos\phi\sin\theta & \cos\theta & \sin\phi\sin\theta \\ -\sin\phi & 0 & \cos\phi \end{pmatrix} \tag{36}$$

$$\mathbf{Q}^\top(\boldsymbol{\omega}) \stackrel{\psi=0}{=} \mathbf{Q}_y^\top(\phi)\mathbf{Q}_z^\top(\theta) = \begin{pmatrix} \cos\phi\cos\theta & \cos\phi\sin\theta & -\sin\phi \\ -\sin\theta & \cos\theta & 0 \\ \sin\phi\cos\theta & \sin\phi\sin\theta & \cos\phi \end{pmatrix} \tag{37}$$

Finally, similar to the 2-dimensional setting, we normalize relative angular positions so that their output range is $[-1, 1)$.

### A.3 Proof of roto-translation invariance

Our method explicitly infers the graph structure over a discrete latent graph and simultaneously learns the dynamical system. Learning the graph structure is a roto-translation invariant task; we want to predict the same edge distribution for each pair of vertices regardless of the global rotation of translation. On the other hand, trajectory forecasting is a roto-translation equivariant task; a global translation and rotation to the input trajectories should affect the output trajectories equivalently. In this section, we derive the proof on roto-translation invariance/equivariance.

Let $\mathbf{Q}_g \in \mathbb{R}^{D \times D}$ be a global rotation matrix in $D$ dimensions and $\boldsymbol{\tau}_g \in \mathbb{R}^{D \times 1}$ be a global translation vector. As explained in section 2.1, input trajectories are described by the states $\mathbf{x}_i^t = [\mathbf{p}_i^t, \mathbf{u}_i^t]$. Similarly, we use $\mathbf{v}_i^t = [\mathbf{p}_i^t, \boldsymbol{\omega}_i^t, \mathbf{u}_i^t]$ to denote the augmented states, described by the linear position, angular position and linear velocity. Finally, we introduce the notation $\mathbf{X}$ and $\mathbf{V}$ to denote the set of states and augmented states, respectively, organized in matrix form.

In the following equations, we remove time indices to reduce clutter. Similar to eq. (3) we define the matrices $\mathbf{R}_g$ and $\tilde{\mathbf{R}}_g$. We have:

$$\mathbf{R}_g = \mathbf{Q}_g \oplus \mathbf{Q}_g \tag{38}$$

$$\tilde{\mathbf{R}}_g = \mathbf{Q}_g \oplus \mathbf{Q}_g \oplus \mathbf{Q}_g \tag{39}$$

Equivalently, we define the augmented translation vectors $\boldsymbol{\delta}_g$ and $\tilde{\boldsymbol{\delta}}_g$:

$$\boldsymbol{\delta}_g = [\boldsymbol{\tau}_g, \mathbf{0}_D] \tag{40}$$

$$\tilde{\boldsymbol{\delta}}_g = [\boldsymbol{\tau}_g, \mathbf{0}_D, \mathbf{0}_D] \tag{41}$$

The definition above holds because velocities and angular positions are translation invariant.

First, we will prove that the transformation to the local coordinate systems is invariant to global translations and rotations. Let J denote the function that converts the augmented states to the local coordinate frames. It is formulated as follows:

$$\mathbf{v}_{j|i} = \mathrm{J}(\mathbf{V})_j \tag{42}$$

$$= \tilde{\mathbf{R}}^\top(\boldsymbol{\omega}_i)[\mathbf{p}_j - \mathbf{p}_i, \boldsymbol{\omega}_j, \mathbf{u}_j] \tag{43}$$

$$= \tilde{\mathbf{R}}^\top(\boldsymbol{\omega}_i)[\mathbf{r}_{j,i}, \boldsymbol{\omega}_j, \mathbf{u}_j] \tag{44}$$

**Local coordinate frames translation invariance**  To prevent the notation from clutter, in the following equations, we will slightly abuse mathematical notation and use the convention $\mathbf{V} + \boldsymbol{\delta}_g$ to denote the translation of each augmented state in $\mathbf{V}$. Programmatically, we can say that we broadcast $\boldsymbol{\delta}_g$ to match the size of $\mathbf{V}$.

$$\mathrm{J}(\mathbf{V} + \boldsymbol{\delta}_g)_j = \tilde{\mathbf{R}}^\top(\boldsymbol{\omega}_i)[\mathbf{p}_j + \boldsymbol{\tau}_g - (\mathbf{p}_i - \boldsymbol{\tau}_g), \boldsymbol{\omega}_j, \mathbf{u}_j] \tag{45}$$

$$= \tilde{\mathbf{R}}^\top(\boldsymbol{\omega}_i)[\mathbf{r}_{j,i}, \boldsymbol{\omega}_j, \mathbf{u}_j] \tag{46}$$

$$= \mathrm{J}(\mathbf{V})_j \tag{47}$$

**Local coordinate frames rotation invariance**   For the canonicalization of the local coordinate systems, we use the matrices $\hat{\mathbf{R}}_i = \hat{\mathbf{R}}(\boldsymbol{\omega}_i)$. These matrices transform under global rotation via the following transformation:

$$\hat{\mathbf{R}}_i = \mathbf{R}_g \cdot \mathbf{R}(\boldsymbol{\omega}_i) \tag{48}$$

$$\hat{\mathbf{R}}_i^\top = \mathbf{R}^\top(\boldsymbol{\omega}_i) \cdot \mathbf{R}_g^\top \tag{49}$$

$$\hat{\tilde{\mathbf{R}}}_i = \tilde{\mathbf{R}}_g \cdot \tilde{\mathbf{R}}(\boldsymbol{\omega}_i) \tag{50}$$

$$\hat{\tilde{\mathbf{R}}}_i^\top = \tilde{\mathbf{R}}^\top(\boldsymbol{\omega}_i) \cdot \tilde{\mathbf{R}}_g^\top \tag{51}$$

Then, we proceed as follows:

$$\mathrm{J}\left(\tilde{\mathbf{R}}_g \cdot \mathbf{V}\right)_j = \hat{\tilde{\mathbf{R}}}^\top(\boldsymbol{\omega}_i) \cdot [\mathbf{Q}_g \cdot \mathbf{p}_j - \mathbf{Q}_g \cdot \mathbf{p}_i, \mathbf{Q}_g \cdot \boldsymbol{\omega}_j, \mathbf{Q}_g \cdot \mathbf{u}_j] \tag{52}$$

$$= \tilde{\mathbf{R}}^\top(\boldsymbol{\omega}_i) \cdot \tilde{\mathbf{R}}_g^\top \cdot [\mathbf{Q}_g \cdot \mathbf{r}_{j,i}, \mathbf{Q}_g \cdot \boldsymbol{\omega}_j, \mathbf{Q}_g \cdot \mathbf{u}_j] \tag{53}$$

$$= \tilde{\mathbf{R}}^\top(\boldsymbol{\omega}_i) \cdot \tilde{\mathbf{R}}_g^\top \cdot \tilde{\mathbf{R}}_g \cdot [\mathbf{r}_{j,i}, \boldsymbol{\omega}_j, \mathbf{u}_j] \tag{54}$$

$$= \tilde{\mathbf{R}}^\top(\boldsymbol{\omega}_i) \cdot [\mathbf{r}_{j,i}, \boldsymbol{\omega}_j, \mathbf{u}_j] \tag{55}$$

$$= \mathrm{J}(\mathbf{V})_j \tag{56}$$

**Encoder roto-translation invariance**   Next, we will prove that the encoder is rotation and translation invariant. Let F denote the encoder. The encoder takes as inputs the set of roto-translated augmented states $\mathbf{V}_{\text{local}} = \{\mathbf{v}_{j|i} \mid j, i \in \{1, \dots, N\}\}$. We have already proven that these inputs are invariant to global translations and rotations. Thus, it follows that the encoder is also roto-translation invariant.

**Decoder roto-translation equivariance**   The decoder takes as inputs the set of roto-translated augmented states $\mathbf{V}_{\text{local}} = \{\mathbf{v}_{j|i} \mid j, i \in \{1, \dots, N\}\}$ as well as the predicted latent edges $\mathbf{z}_{j,i}$. We use $\mathbf{Z} = \{\mathbf{z}_{j,i} \mid j, i \in \{1, \dots, N\}, j \neq i\}$ to denote the set of all latent edges.

To prove that the decoder is equivariant to global rotations and translations, we will split its functionality into 2 consecutive components. Let G be the first component that predicts the differences in position and velocity $\boldsymbol{\Delta}\mathbf{x}$ in the local coordinate systems. Let H be the second component that transforms the predictions from the local coordinate systems to the global coordinate system, as described by eq. (70). The first part of the decoder takes as inputs the augmented states $\mathbf{V}_{\text{local}}$ as well as the latent edges $\mathbf{Z}$. $\mathbf{Z}$ is the output of the encoder, and as we proved earlier, it is invariant. $\mathbf{V}_{\text{local}}$ is also invariant. Hence, G is roto-translation invariant.

Finally, we have to prove that H is equivariant to global translations and rotations. H is a function of $\mathbf{X}$ and $\mathbf{V}_{\text{local}}$ and is defined as $\mathrm{H}(\mathbf{X}, \mathbf{V}_{\text{local}})_i = \mathbf{x}_i + \mathbf{R}(\boldsymbol{\omega}_i) \cdot \mathrm{G}(\mathbf{V}_{\text{local}})_i$.

First, we will prove that H is translation equivariant. We have the following:

$$\mathrm{H}(\mathbf{X}, \mathbf{V}_{\text{local}})_i = \mathbf{x}_i + \mathbf{R}(\boldsymbol{\omega}_i) \cdot \mathrm{G}(\mathbf{V}_{\text{local}})_i \tag{57}$$

$$\mathrm{H}\left(\mathbf{X} + \boldsymbol{\delta}_g, \mathbf{V}_{\text{local}} + \tilde{\boldsymbol{\delta}}_g\right)_i = \mathbf{x}_i + \boldsymbol{\delta}_g + \mathbf{R}(\boldsymbol{\omega}_i) \cdot \mathrm{G}\left(\mathbf{V}_{\text{local}} + \tilde{\boldsymbol{\delta}}_g\right)_i \tag{58}$$

$$= \mathbf{x}_i + \boldsymbol{\delta}_g + \mathbf{R}(\boldsymbol{\omega}_i) \cdot \mathrm{G}(\mathbf{V}_{\text{local}})_i \tag{59}$$

$$= \mathrm{H}(\mathbf{X}, \mathbf{V}_{\text{local}})_i + \boldsymbol{\delta}_g \tag{60}$$

Next, we will prove that H is rotation equivariant. We have the following:

$$\mathrm{H}\left(\mathbf{R}_g \cdot \mathbf{X}, \tilde{\mathbf{R}}_g \cdot \mathbf{V}_{\text{local}}\right)_i = \mathbf{R}_g \cdot \mathbf{x}_i + \mathbf{R}_g \cdot \mathbf{R}(\boldsymbol{\omega}_i) \cdot \mathrm{G}\left(\tilde{\mathbf{R}}_g \cdot \mathbf{V}_{\text{local}}\right)_i \tag{61}$$

$$= \mathbf{R}_g \cdot (\mathbf{x}_i + \mathbf{R}(\boldsymbol{\omega}_i) \cdot \mathrm{G}(\mathbf{V}_{\text{local}})_i) \tag{62}$$

$$= \mathbf{R}_g \cdot \mathrm{H}(\mathbf{X}, \mathbf{V}_{\text{local}})_i \tag{63}$$

# B Implementation details

## B.1 LoCS

**Encoder & Prior**  The embeddings $\mathbf{h}_{j,i}^{(2)}$ are fed into 2 LSTMs [22], one forward in time that computes the prior and one backwards in time for the encoder. The hidden state from the forward LSTM is used to compute the prior distribution, while the hidden states from both the forward and the backward LSTM are concatenated to compute the encoder distribution, according to the following equations:

$$\mathbf{h}_{(j,i),\text{prior}}^t = \text{LSTM}_{\text{prior}}\left(\mathbf{h}_{j,i}^{(2)}, \mathbf{h}_{(j,i),\text{prior}}^{t-1}\right) \tag{64}$$

$$\mathbf{h}_{(j,i),\text{enc}}^t = \text{LSTM}_{\text{enc}}\left(\mathbf{h}_{j,i}^{(2)}, \mathbf{h}_{(j,i),\text{enc}}^{t+1}\right) \tag{65}$$

$$p_\phi\left(\mathbf{z}^t | \mathbf{x}^{1:t}, \mathbf{z}^{1:t-1}\right) = \text{softmax}\left(f_{\text{prior}}\left(\mathbf{h}_{(j,i),\text{prior}}^t\right)\right) \tag{66}$$

$$q_\phi\left(\mathbf{z}_{j,i}^t | \mathbf{x}\right) = \text{softmax}\left(f_{\text{enc}}\left(\left[\mathbf{h}_{(j,i),\text{prior}}^t, \mathbf{h}_{(j,i),\text{enc}}^t\right]\right)\right) \tag{67}$$

The functions $f_{\text{enc}}, f_{\text{prior}}$ are MLPs that map the hidden states to $\mathbb{R}^K$, where $K$ is the number of latent edge types.

**Decoder**  Following [19, 27], we use 2 different decoders based on whether the governing dynamics are Markovian. In both cases, the decoders have similar structure with [19, 27]; the main difference is that we operate entirely on the roto-translated local coordinate frames. In order to convert our predictions back to the global coordinate frame, we perform an inverse rotation by $\mathbf{R}_i^t = \mathbf{R}(\boldsymbol{\omega}_i^t) = \mathbf{Q}(\boldsymbol{\omega}_i^t) \oplus \mathbf{Q}(\boldsymbol{\omega}_i^t)$.

**Markovian decoder**  In many applications, such as dynamical systems in physics, the governing dynamics satisfy the Markov property $p_\theta(\mathbf{x}^{t+1} | \mathbf{x}^{1:t}, \mathbf{z}^{1:t}) = p_\theta(\mathbf{x}^{t+1} | \mathbf{x}^t, \mathbf{z}^t)$. In this case, we use the following decoder:

$$\mathbf{m}_{j,i}^t = \sum_k z_{(j,i),k}^t f^k\left(\left[\mathbf{v}_{j|i}^t, \mathbf{v}_{i|i}^t\right]\right) \tag{68}$$

$$\mathbf{m}_i^t = f_v^{(3)}\left(g_v^{(3)}\left(\mathbf{v}_{i|i}^t\right) + \frac{1}{|\mathcal{N}(i)|}\sum_{j \in \mathcal{N}(i)} \mathbf{m}_{j,i}^t\right) \tag{69}$$

$$\boldsymbol{\mu}_i^{t+1} = \mathbf{x}_i^t + \mathbf{R}_i^t \cdot f_v^{(4)}\left(\mathbf{m}_i^t\right) \tag{70}$$

$$p(\mathbf{x}_i^{t+1} | \mathbf{x}^t, \mathbf{z}^t) = \mathcal{N}\left(\boldsymbol{\mu}_i^{t+1}, \sigma^2 \mathbf{I}\right) \tag{71}$$

The functions $f_v^{(3)}, f_v^{(4)}$ and $f^k, k \in \{1, \ldots K\}$ are MLPs, while $g_v^{(3)}$ is a linear layer. The output of the model is the mean estimate of a multivariate isotropic Gaussian distribution with fixed variance.

**Recurrent decoder**  In most real-world applications, the Markovian assumption does not hold. In this case we use a recurrent decoder.

$$\mathbf{m}_{j,i}^t = \sum_k z_{(j,i),k}^t f^k\left(\left[\mathbf{v}_{j|i}^t, \mathbf{v}_{i|i}^t\right]\right) \tag{72}$$

$$\mathbf{m}_i^t = f_v^{(3)}\left(g_v^{(3)}\left(\mathbf{v}_{i|i}^t\right) + \frac{1}{|\mathcal{N}(i)|}\sum_{j \in \mathcal{N}(i)} \mathbf{m}_{j,i}^t\right) \tag{73}$$

$$\mathbf{h}_{j,i}^t = \sum_k z_{(j,i),k}^t g^k\left(\left[\mathbf{h}_j^t, \mathbf{h}_i^t\right]\right) \tag{74}$$

$$\mathbf{n}_i^t = \frac{1}{|\mathcal{N}(i)|}\sum_{j \in \mathcal{N}(i)} \mathbf{h}_{(j,i)}^t \tag{75}$$

$$\mathbf{h}_i^{t+1} = \text{GRU}\left(\left[\mathbf{n}_i^t, \mathbf{m}^t\right], \mathbf{h}_i^t\right) \tag{76}$$

$$\boldsymbol{\mu}_i^{t+1} = \mathbf{x}_i^t + \mathbf{R}_i^t \cdot f_v^{(4)}\big(\mathbf{h}_i^{t+1}\big) \tag{77}$$

$$p(\mathbf{x}_i^{t+1}|\mathbf{x}^{1:t}, \mathbf{z}^{1:t}) = \mathcal{N}\big(\boldsymbol{\mu}_i^{t+1}, \sigma^2\mathbf{I}\big) \tag{78}$$

The functions $g^k, k \in \{1, \dots K\}$ are MLPs. The GRU block [7] is identical to the one used in [27].

**Loss**  Following [19], we train our models by minimizing the Evidence Lower Bound (ELBO), which comprises the reconstruction loss of the predicted trajectories (positions and velocities) and the KL divergence.

$$\mathcal{L}(\phi, \theta) = \mathbb{E}_{q_\phi(\mathbf{z}|\mathbf{x})}[\log p_\theta(\mathbf{x}|\mathbf{z})] - \mathrm{KL}[q_\phi(\mathbf{z}|\mathbf{x})||p_\phi(\mathbf{z}|\mathbf{x})] \tag{79}$$

As mentioned earlier, we assume the outputs follow an isotropic Gaussian distribution with fixed variance. The reconstruction loss and the KL divergence take the following form:

$$\mathbb{E}_{q_\phi(\mathbf{z}|\mathbf{x})}[\log p_\theta(\mathbf{x}|\mathbf{z})] = -\sum_i \sum_t \frac{||\mathbf{x}_i^t - \boldsymbol{\mu}_i^t||}{2\sigma^2} + \frac{1}{2}\log\big(2\pi\sigma^2\big) \tag{80}$$

$$\mathrm{KL}[q_\phi(\mathbf{z}|\mathbf{x})||p_\phi(\mathbf{z}|\mathbf{x})] = \sum_{t=1}^T \left( \mathbb{H}(q_\phi(\mathbf{z}_{ji}^t|\mathbf{x})) - \sum_{\mathbf{z}_{ji}^t} q_\phi(\mathbf{z}_{ji}^t|\mathbf{x}) \log p_\theta(\mathbf{z}_{ji}^t|\mathbf{x}^{1:t}, \mathbf{z}^{1:t-1}) \right) \tag{81}$$

$\mathbb{H}$ denotes the entropy operator. In all experiments, we set $\sigma^2 = 10^{-5}$.

**Spherical coordinate relative positions**  Many works in the literature [41, 44] employ distance-based message-passing steps and filters as a means to better model interactions. We also find that explicitly incorporating Euclidean distances is useful in practice. We augment the canonicalized states $\mathbf{v}_{j|i}$ with the spherical representations of the relative positions $\mathbf{r}_{j,i}$. We denote the spherical relative positions as $\mathbf{s}_{j,i}$. They are computed as follows:

$$\mathbf{s}_{j,i}^t = \mathrm{cart2spherical}\big(\mathbf{Q}^{t\top}(\boldsymbol{\omega}_i) \cdot \mathbf{r}_{j,i}^t\big) \tag{82}$$

The spherical representations are computed within the roto-translated coordinate frames and thus, have no effect on the roto-translation invariance. We modify eq. (6), omitting the time indices for clarity:

$$\mathbf{h}_{j,i}^{(1)} = f_e^{(1)}\big(\big[\mathbf{v}_{j|i}, \mathbf{s}_{j,i}, \mathbf{v}_{i|i}\big]\big) \tag{83}$$

Similarly, we modify eqs. (68) and (72) as follows:

$$\mathbf{m}_{j,i}^t = \sum_k z_{(j,i),k}^t f^k\left(\left[\mathbf{v}_{j|i}^t, \mathbf{s}_{j,i}^t, \mathbf{v}_{i|i}^t\right]\right) \tag{84}$$

**Anisotropic filtering**  The anisotropic filters presented in section 3.4 are used to compute the latent edge embeddings. Specifically, we replace the filters in eq. (6) in the encoder. The filter generating network is a 2-layer MLP with ELU [9] activation in the hidden layer. In the inD [4] experiment, we also use anisotropic filters in the decoder. These filters replace the filters in eq. (72). In this case, we use a 2-layer MLP with $\mathtt{tanh}$ activation in the hidden layer. In all experiments, we use the spherical relative positions as input to the filter generating network instead of the Cartesian relative positions. Using $\boldsymbol{\Delta}\mathbf{p}_{j,i}^t = \big[\mathbf{s}_{j,i}^t, \mathbf{Q}^{t\top}(\boldsymbol{\omega}_i) \cdot \boldsymbol{\omega}_j^t\big]$ to denote the canonicalized relative linear and angular positions, the encoder and decoder filter generating networks are formulated as:

$$\mathbf{h}_{j,i}^{(1),t} = \mathbf{W}_{\mathcal{F}}\big(\boldsymbol{\Delta}\mathbf{p}_{j,i}^t\big) \cdot \left[\mathbf{v}_{j|i}^t, \mathbf{s}_{j,i}^t, \mathbf{v}_{i|i}^t\right] \tag{85}$$

$$\mathbf{m}_{j,i}^t = \sum_k z_{(j,i),k}^t \mathbf{W}_{\mathcal{F}}^k\big(\boldsymbol{\Delta}\mathbf{p}_{j,i}^t\big) \cdot \left(\left[\mathbf{v}_{j|i}^t, \mathbf{s}_{j,i}^t, \mathbf{v}_{i|i}^t\right]\right) \tag{86}$$

## B.2  NRI & dNRI

We use the official dNRI implementation from `https://github.com/cgraber/cvpr_dNRI`. We use the same repository for its NRI implementation as well.

## B.3 EGNN

We use the official EGNN implementation from `https://github.com/vgsatorras/egnn`. In all experiments we use the EGNN model with position and velocity inputs/outputs. Each layer is defined as $\mathbf{h}^{l+1}, \mathbf{p}^{l+1}, \mathbf{u}^{l+1} = \text{EGCL}(\mathbf{h}^l, \mathbf{p}^l, \mathbf{u}^l)$.

The hidden state at the input layer $\mathbf{h}_i^0$ is computed via a linear layer $\psi_h$ that embeds the input (scalar) speed of each node to the hidden dimension of the model, $\mathbf{h}_i^0 = \psi_h(\|\mathbf{u}_i^0\|)$. Each EGNN layer is formulated as follows:

$$\mathbf{m}_{j,i} = \phi_e\left(\mathbf{h}_i^l, \mathbf{h}_j^l, \|\mathbf{p}_j^l - \mathbf{p}_i^l\|_2^2\right) \tag{87}$$

$$\mathbf{u}_i^{l+1} = \phi_v(\mathbf{h}_i^l)\mathbf{u}_i^l + \frac{1}{|\mathcal{N}(i)|}\sum_{j\in\mathcal{N}(i)}(\mathbf{p}_j^l - \mathbf{p}_i^l)\cdot\phi_x(\mathbf{m}_{j,i}) \tag{88}$$

$$\mathbf{p}_i^{l+1} = \mathbf{p}_i^l + \mathbf{u}_i^{l+1} \tag{89}$$

$$\mathbf{m}_i = \sum_{j\in\mathcal{N}(i)}\phi_w(\mathbf{m}_{j,i})\cdot\mathbf{m}_{j,i} \tag{90}$$

$$\mathbf{h}_i^{l+1} = \phi_h(\mathbf{h}_i^l, \mathbf{m}_i) \tag{91}$$

The functions $\phi_e, \phi_v, \phi_x, \phi_h, \phi_w$ are MLPs with learnable parameters that closely follow the original work. More specifically, the functions $\phi_e, \phi_v, \phi_x$ and $\phi_h$ are 2-layer MLPs, and the function $\phi_w$ is a linear layer with a sigmoid activation used to weigh the messages before aggregation.

The EGNN model comprises 4 layers and the hidden dimensions in all layers are 64. The training and evaluation schemes are identical to the other models, except that the model is trained by minimizing the negative log-likelihood of a Gaussian distribution of the positions and velocities, following Equation 80.

## B.4 Computing resources

We ran all experiments on internal clusters using single GPU jobs. 3 different GPU models were used in total, namely the Nvidia RTX 2080 Ti, Nvidia GTX 1080 Ti, and Nvidia TitanX. The source code was written in PyTorch [61], version 1.4.0 with CUDA 10.0.

## B.5 Hyperparameters & training details

For the synthetic experiment, we follow [19] and train models for 200 epochs. We use 2 edges types and hardcode the first edge type to indicate absence of interactions, with a no-edge prior of 0.9. For the charged particles [27], we use 2 edges types with a uniform prior and train the models for 200 epochs. For inD [4], we follow [19] and train models for 400 epochs. We use 4 edge types and hardcode the first to indicate absence of interactions. For motion capture [10] subject #35, we follow [19] and train models for 600 epochs. We use 4 edge types and hardcode the first to indicate absence of interactions. In all experiments, we train LoCS using Adam [25] with a learning rate of $5e-4$.

**Encoder & Prior** $f_v^{(1)}$ and $f_e^{(2)}$ are 2-layer MLPs with ELU [9] activations after each layer and Batch Normalization [58] at the end, with 256 hidden and output dimensions. $g_v^{(1)}$ is a linear layer with 256 output dimensions. $\text{LSTM}_{\text{prior}}$ and $\text{LSTM}_{\text{enc}}$ are LSTMs [22] with 64 hidden dimensions. $f_{\text{prior}}$ and $f_{\text{enc}}$ are 3-layer MLPs with ELU activations after the first 2 layers, 128 hidden dimensions, and $K$ output dimensions, where $K$ is the number of latent edge types. The filter generating network $\mathbf{W}_{\mathcal{F}}$ is a 2-layer MLP, with ELU after the first layer, 256 hidden dimensions. For the experiment on inD [4], we use 64 hidden dimensions for the filter generating network instead.

**Decoder** $g_v^{(3)}$ is a linear layer with 256 output dimensions. $f_k$ are 2-layer MLPs with ReLU [60] activations after each layer and $g_k$ are 2-layer MLPs with $\tanh$ activations after each layer. $f_v^{(3)}$ is the identity function and $f_v^{(4)}$ is a 3-layer MLP with ReLU activations after the first 2 layers, 256 hidden dimensions and $2D$ output dimensions.

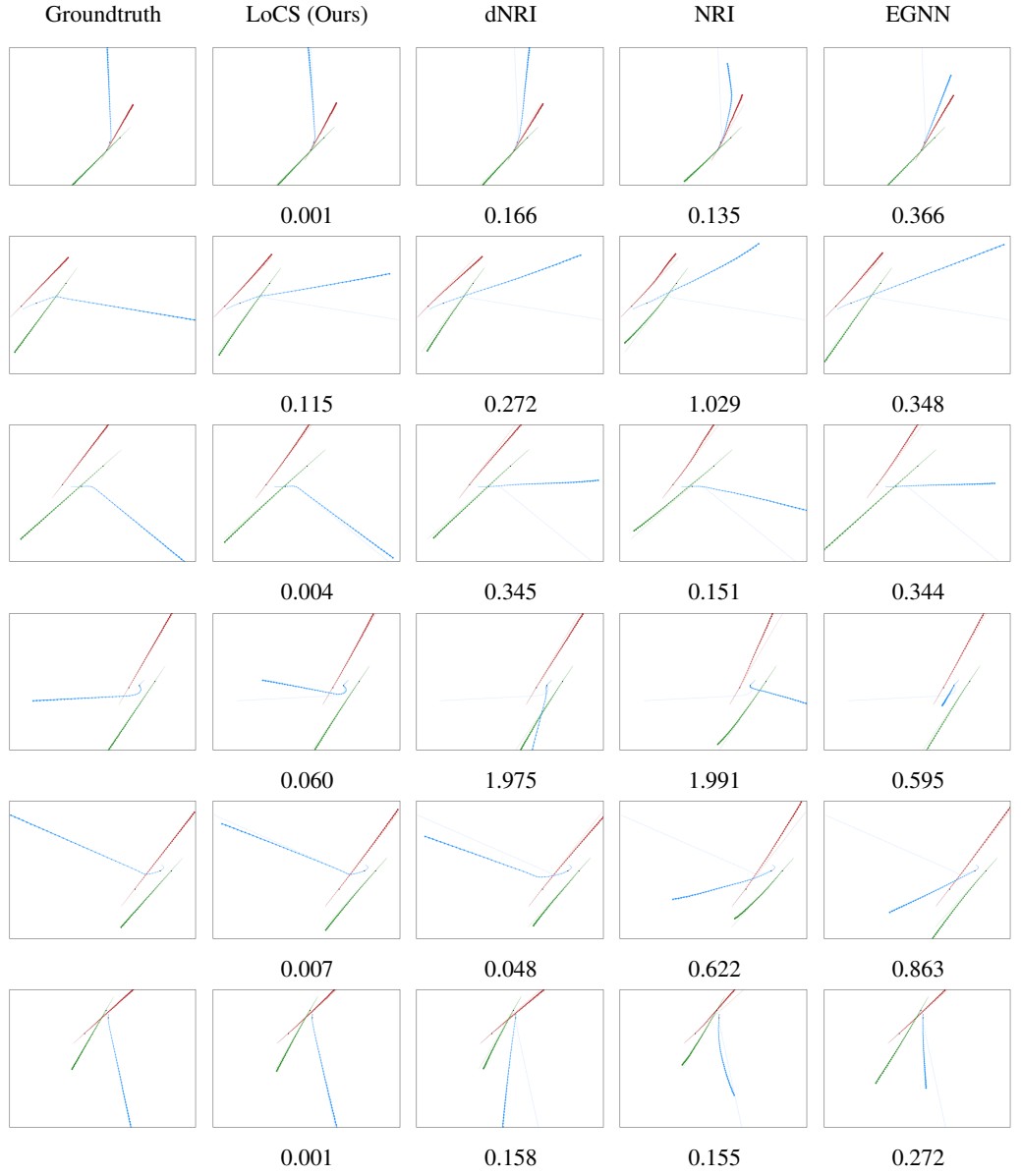

Figure 7: Qualitative results on synthetic dataset, scenes #0 – #5

The filter generating network $\mathbf{W}_{\mathcal{F}}$ is a 2-layer MLP, with tanh activation after the first layer and 256 hidden dimensions. For the experiment on inD [4], we use 64 hidden dimensions for the filter generating network instead.

The GRU block [7] in the recurrent decoder is identical to the one used in [27], with 256 hidden dimensions.

## C   Qualitative results

### C.1   Synthetic

Figure 7 shows comparative qualitative results for the synthetic dataset [19]. The numbers below each sub-figure represent respective MSE errors.

### C.2 Charged

Figure 8 shows comparative qualitative results for 3D charged particles [27]. The numbers below each sub-figure represent respective errors.

### C.3 InD

Figure 9 shows comparative qualitative results for inD [4].

### C.4 Charged - Interactive

Figure 10 shows comparative qualitative results for the highly interactive subset of 3D charged particles. The numbers below each sub-figure represent respective errors.

## D Quantitative results

### D.1 Charged particles

Figure 11 shows MSE and $L_2$ errors for charged particles [27].

### D.2 Traffic trajectory forecasting

Figure 12 shows MSE and $L_2$ errors for inD [4].

### D.3 Motion capture

Figure 13 shows MSE and $L_2$ errors for motion capture [10], subject #35.

### D.4 Ablation experiments

The following figures show the complete error curves for the ablation experiments. Figure 14 shows the errors for the highly interactive charged particles subset. Figure 15 shows the results of training dNRI using speed normalization. Figure 16 shows the impact of anisotropic continuous filtering in our method. The roto-translated local coordinate frames already outperform compared methods, while incorporating the anisotropic filters boosts performance even further. Finally, fig. 17 shows the impact of rotation in local coordinate frames, specifically in scenarios without intrinsic orientations, such as charged particles.


Figure 8: Qualitative results on charged particles, scenes #0 − #5

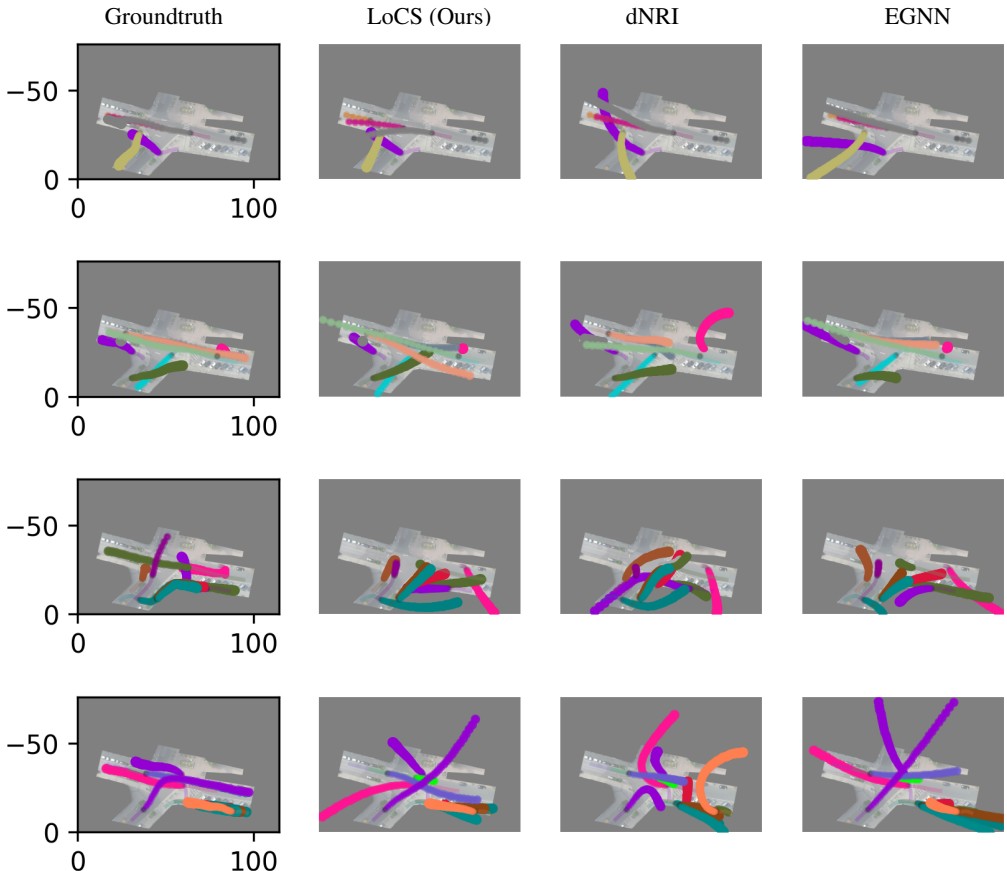

Figure 9: Qualitative results on inD

    (c) Did you report error bars (e.g., with respect to the random seed after running experiments multiple times)? [Yes]

    (d) Did you include the total amount of compute and the type of resources used (e.g., type of GPUs, internal cluster, or cloud provider)? [Yes] See appendix B.4

4. If you are using existing assets (e.g., code, data, models) or curating/releasing new assets...

    (a) If your work uses existing assets, did you cite the creators? [Yes]

    (b) Did you mention the license of the assets? [Yes]

    (c) Did you include any new assets either in the supplemental material or as a URL? [No] Code will be released upon acceptance

    (d) Did you discuss whether and how consent was obtained from people whose data you're using/curating? [Yes]

    (e) Did you discuss whether the data you are using/curating contains personally identifiable information or offensive content? [N/A]

5. If you used crowdsourcing or conducted research with human subjects...

    (a) Did you include the full text of instructions given to participants and screenshots, if applicable? [N/A]

    (b) Did you describe any potential participant risks, with links to Institutional Review Board (IRB) approvals, if applicable? [N/A]

    (c) Did you include the estimated hourly wage paid to participants and the total amount spent on participant compensation? [N/A]

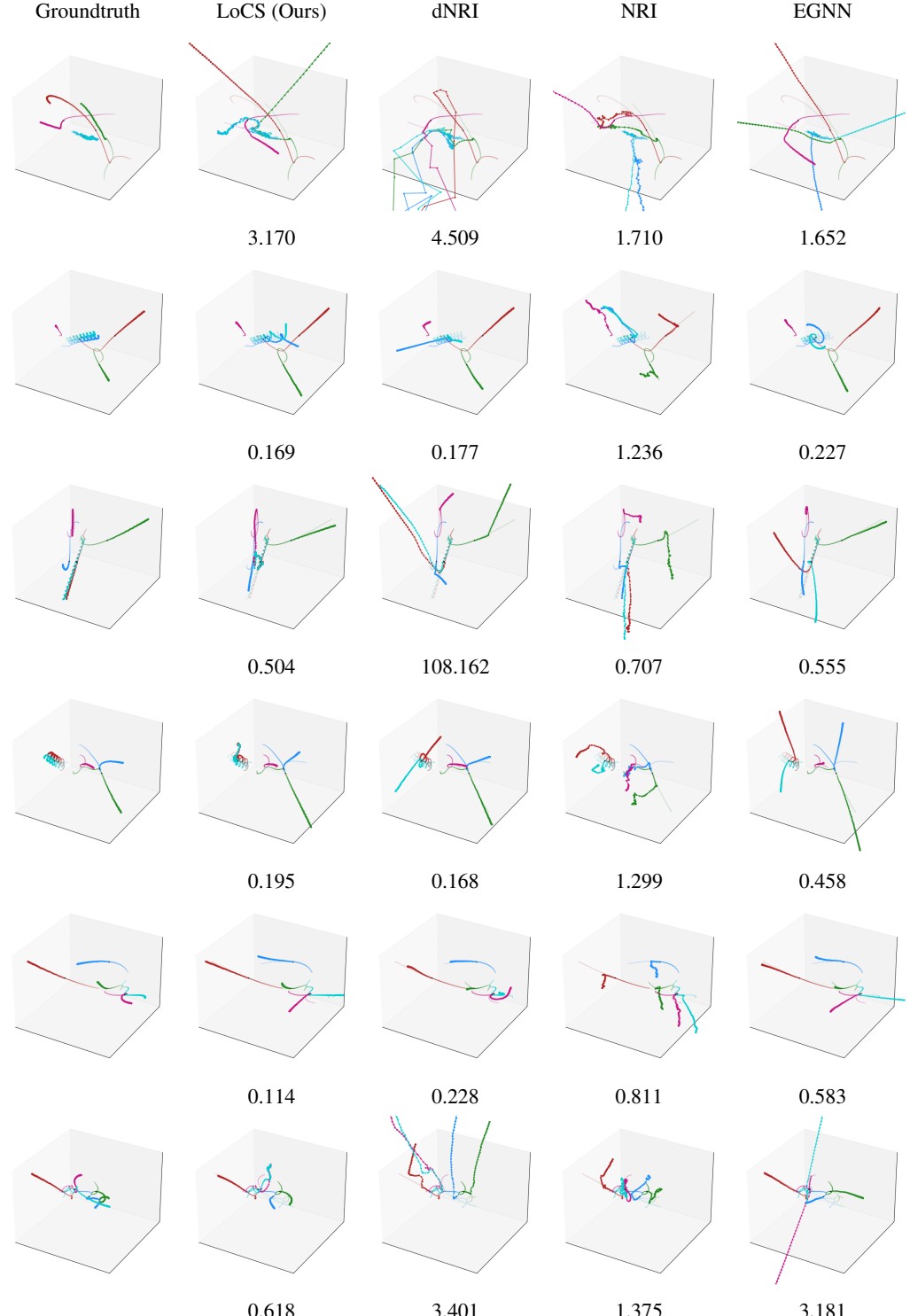

Figure 10: Qualitative results on interactive subset of charged particles, scenes #0 – #5

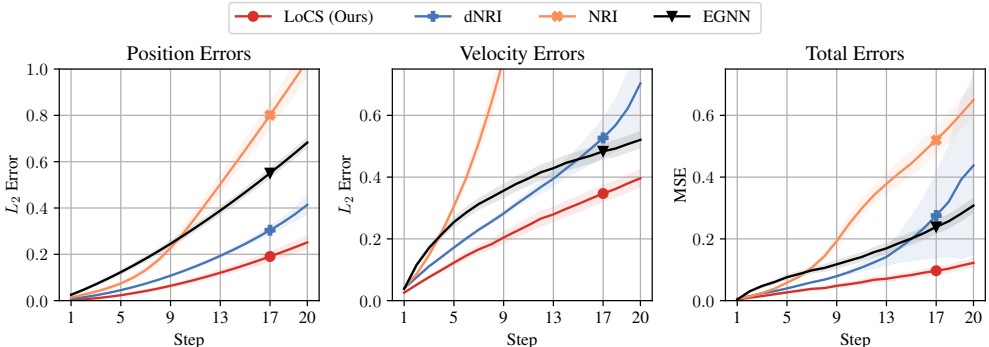

Figure 11: Results on Charged particles dataset

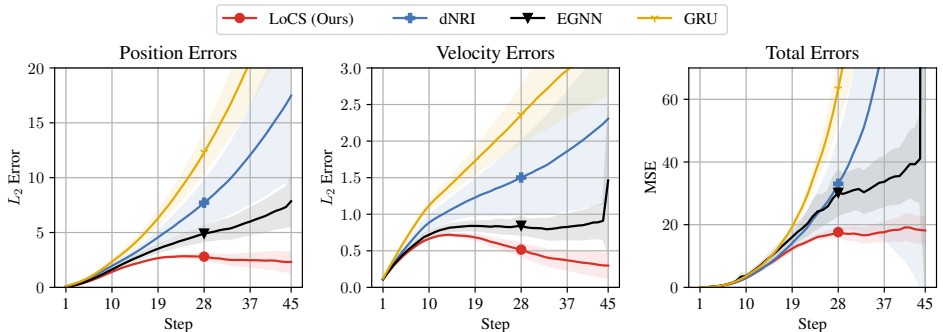

Figure 12: Results on InD dataset

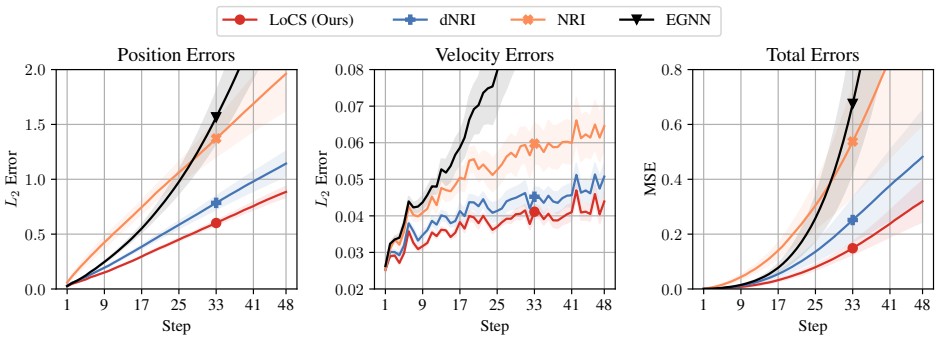

Figure 13: Results on motion capture (#35)

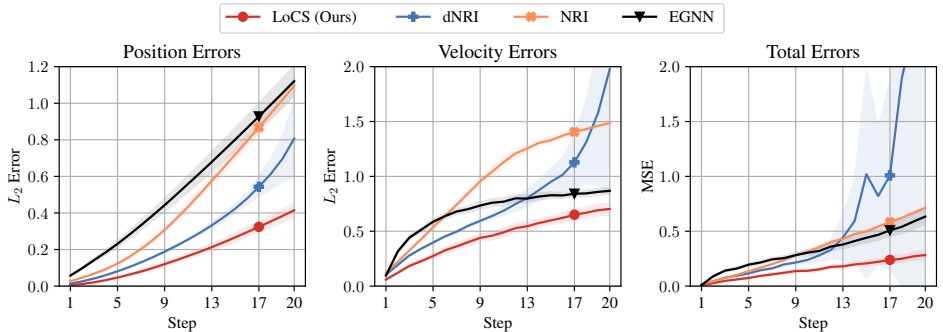

Figure 14: Results on Charged particles interactive subset

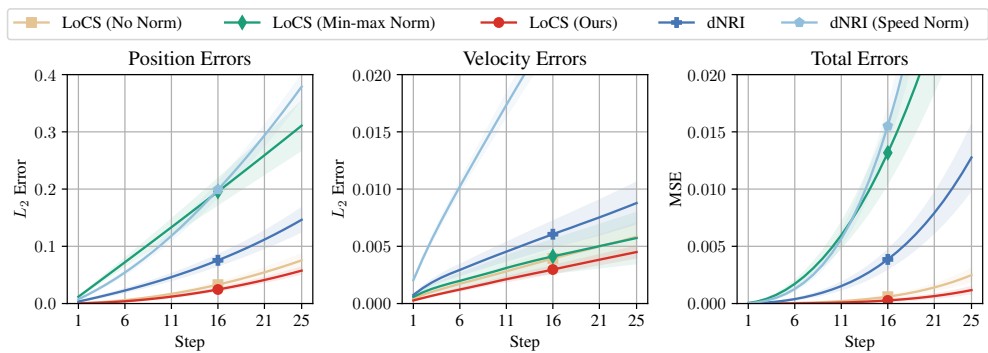

Figure 15: Results on synthetic dataset; impact of speed norm normalization

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

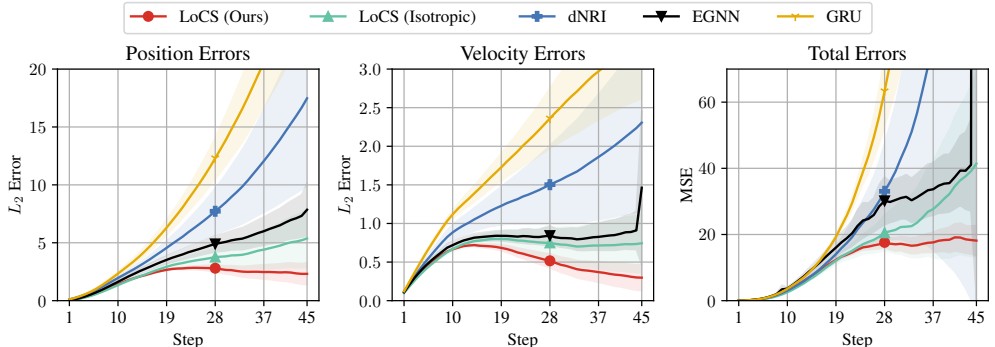

Figure 16: Results on InD dataset; impact of anisotropic filtering

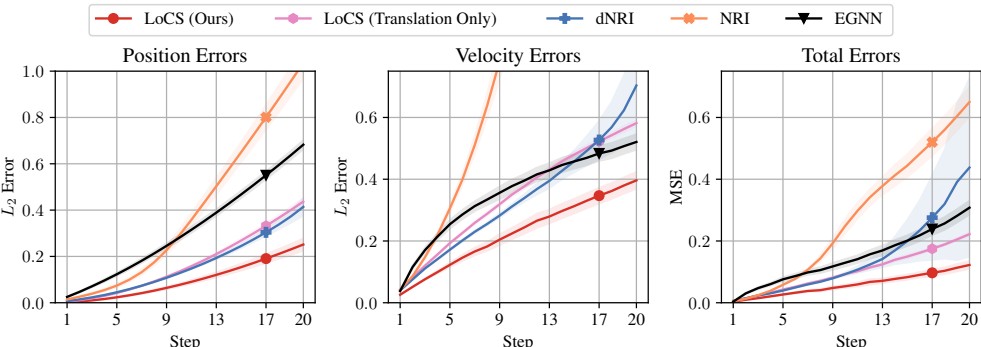

Figure 17: Results on charged particles dataset; impact of rotation in roto-translated local coordinate frames