# OpenReview forum: "Roto-translated Local Coordinate Frames For Interacting Dynamical Systems"
_NeurIPS.cc/2021/Conference — NeurIPS 2021 Poster_

### Official Review · Reviewer_fJBY · 2021-07-08

**Rating:** 7
**Confidence:** 2

**Summary:**

This paper presents an approach to  enable roto-translation invariant edge prediction and roto-translation equivariant trajectory forecasting. Rotation and translation invariants are achieved by transforming global coordinate into local coordinate frames of each object in the graph neural network. Experiments are done on forecasting trajectories of charged particles, traffics and human motions.

**Ethical Concerns:**

None.

**Limitations And Societal Impact:**

Yes, the authors address the limitation.

**Main Review:**

Pro:

1. The idea of representing neighbors coordinate in its own local coordinate frame is simple and straightforward. And the improvement seems substantial in the experiments presented.

2. Detailed derivation provided to support the soundness of the method.

Questions

1. It is not clear to me how you get the angular position approximation when the velocity is 0.

2. I am also not sure how you can represent full 3D orientation using euler angle only since it will encounter singularity at some point.

3. Figure 3 doesn't really tell us anything without the ground truth. Even though more data are provided in the appendix, figure 3 seems redundant.

4. I don't follow why we are doing a speed normalization ablation on dNRI, shouldn't it perform on LoCS? i.e, compare LoCS with or without speed normalization.

5. What does GRU in Figure 4b, Figure 5c refer to?

**Time Spent Reviewing:**

3

---

> ### Author Response · Authors · 2021-08-10
> **Author Response to Reviewer fJBY**
>
> We thank the reviewer for the encouraging words and positive feedback. Please find our answers below.
>
> **It is not clear to me how you get the angular position approximation when the velocity is 0.**
>
> The orientation of non-moving objects is set to be 0.
> While this approximation is arbitrary and posits a limitation of our method, it does not seem to hinder the performance of our method.
> We attribute this to the fact that often times, the orientations of neighbouring non-moving objects carry low inductive information, at least in the datasets available to us.
>
> **I am also not sure how you can represent full 3D orientation using euler angle only since it will encounter singularity at some point.**
>
> Euler angles are indeed prone to singularities and gimbal lock.
> As with other methods relying on similar geometric rotations, local coordinate frames could somehow be adversely affected as well.
> In practice, we never experienced any issues with training or evaluating the method in any of our experiments, nor did we observe predicted trajectories that would point to singularities or gimbal lock.
> That is probably because in the end, we only use Euler angles as part of the feature vectors, which are then fed to the graph neural network; we did not use the angles as a representation itself.
> That said, it would be very interesting to investigate different orientation representations that do not suffer from singularities, such as quaternions.
>
> **Figure 3 doesn't really tell us anything without the ground truth. Even though more data are provided in the appendix, figure 3 seems redundant.**
>
> The groundtruth trajectories are plotted with semi-transparent colors, which arguably appear almost invisible.
> We apologize and will fix them for the camera-ready version.
>
> **I don't follow why we are doing a speed normalization ablation on dNRI, shouldn't it perform on LoCS? i.e, compare LoCS with or without speed normalization.**
>
> The ablation study was intended to show that speed normalization is not the main cause for the improvements for LoCS.
> If it were, it should also benefit dNRI.
> To remove any doubts, as suggested we perform an extra ablation for LoCS with and without speed normalization.
> Results are shown in the table below.
> We observe that when the inputs are normalized using min-max normalization, LoCS underperforms especially for predicting future positions. Without any normalization, LoCS already performs better than other baselines.
> Using speed normalization improves the accuracy of LoCS even further.
>
> We conclude that LoCS attain the highest accuracies when combined with speed normalization, although it works quite well even without it.
>
> Method|MSE@t=1          |MSE@t=12         |MSE@t=25         |
> -----:|-------          |--------         |--------         |
> LoCS (Speed Norm)   |2.84e-07±1.91e-07|1.13e-04±5.35e-05|1.17e-03±2.96e-04|
> LoCS (No Norm)      |7.53e-07±3.88e-07|2.61e-04±8.32e-05|2.49e-03±5.74e-04|
> LoCS (Min-max Norm) |4.64e-05±6.24e-06|7.16e-03±1.34e-03|3.46e-02±8.03e-03|
> dNRI                |1.03e-05±1.67e-06|1.88e-03±4.10e-04|1.28e-02±2.86e-03|
> NRI                 |5.39e-05±1.83e-05|1.04e-02±3.46e-03|7.96e-02±2.98e-02|
> EGNN                |3.67e-06±5.03e-08|2.16e-03±1.90e-04|1.40e-02±3.29e-03|
>
> **What does GRU in Figure 4b, Figure 5c refer to?**
> GRU refers to a Gated Recurrent unit [Cho et al., 2014], a non-interactive recurrent network used as a baseline.
> We will update the text to make the reference explicit.
>
> References
> ---
>
> Cho, Kyunghyun, Bart Van Merriënboer, Caglar Gulcehre, Dzmitry Bahdanau, Fethi Bougares, Holger Schwenk, and Yoshua Bengio. "Learning phrase representations using RNN encoder-decoder for statistical machine translation." arXiv preprint arXiv:1406.1078 (2014).

---

### Official Review · Reviewer_49Zp · 2021-07-14

**Rating:** 7
**Confidence:** 2

**Summary:**

This paper investigates the problem of simultaneous learning of interactions and dynamics of dynamical systems of multiple interacting objects. The proposed method is built upon dynamics NRI and incorporate a design Roto-translated invariance. The key of the Roto-translated invariance design is a local coordinate frame design - one for each object. The feature vectors that are fed into GNN are based on local coordinate frame representations instead of global coordinate frame representations. The results shows a clear advantage over previous methods.

**Limitations And Societal Impact:**

Yes, the authors have addressed the limitations. Societal impact doesn't apply in this case.

**Main Review:**

The paper leverages symmetries and built it into the computation graph to improve the learning of interactions and dynamics. The idea is straight forward, but the writing of the paper need to be improved to make it easier for readers to appreciate it. For example, Figure 1 is very helpful for people to understand the high-level idea, but it is too far away from the presentation of the idea (Section 3.1) and it is not referred to in the text. The authors refer to Figure 6 in Section 5.2 and Figure 7 in Section 5.3, but those figures are not in the main text. It seems they are not referring to Figure 6 and 7 in the supplementary materials either.

I have one key question: Your augmented state only consists of linear position, angular position and linear velocity. You are not including angular velocity. I guess this is because you assume they are point mass and cannot rotate. In your examples, the particles indeed cannot rotate. If this is the case, I wonder why including the angular position and the rotation transformation is necessary. In other words, what if you only have the translation transformation to form each local frame? In your framework, this reduces to the case that you don't have angular position data and you approximate the angular positions as the global x direction instead of the direction of velocity vector. I think it would be good to add it as an ablation study so that we know the importance of rotation translation. Intuitively I can't think of any reason rotation transformation would help if you only have particles instead of 3D rigid bodies.

One nitpik: Figure 1 is a little misleading since the visual orientation of the cartoons are fixed across figures. For example, the polygon is visually at the same orientation even though the arrow indicates the orientations should be different. It's easier to change them all to circles but I guess the authors use different shapes to be color-blind friendly.


**Time Spent Reviewing:**

5

---

> ### Author Response · Authors · 2021-08-10
> **Author Response to Reviewer 49Zp**
>
> We thank the reviewer for the encouraging words and positive feedback. Please find our answers below.
>
> **Your augmented state only consists of linear position, angular position and linear velocity. You are not including angular velocity. I guess this is because you assume they are point mass and cannot rotate.**
>
> Angular velocities may be included in the augmented state if we know the rate of change of orientation.
> However, angular velocities are not always available.
> This can be either because angular velocities are simply not recorded in the datasets available to us (e.g., the inD dataset [Bock et al., 2019] does not include the angular velocity of the cars).
> Or, this can be because the objects in the experimental setting are seen as point masses (e.g., in the charged particle dataset [Kipf et al., 2018]), in which case an angular velocity does not make sense to begin with.
>
> When this information is not available, we propose to approximate angular velocities using the angles of the acceleration vectors.
> Replacing the unavailable angular velocities with angles of acceleration vectors works just as well because -in the end- we mainly care to have a consistent (invariant) direction with which to define the local coordinate frame across time steps.
>
> **Intuitively I can't think of any reason rotation transformation would help if you only have particles instead of 3D rigid bodies.**
>
> Even though particles do not have intrinsic orientations, we postulate that roto-translated local coordinate frames allow for more efficient learning, as long as they are consistently invariant.
> To motivate this, consider the simple case of a 2D system on paper with three objects A, B, C with no intrinsic orientation, laid out like
>
> ```
> B
>
>
>   A (0,0)
>      C
> ```
>
> In this case, B is far upper left compared to A, while C is near below right compared to A.
> Consider now rotating the system by $\pi$ as in
>
> ```
> C
>    A (0,0)
>
>
>      B
> ```
>
> In both cases A lies at the same origin, however, now C is near upper left and B is far below right.
> If we do not canonicalize with respect to rotations, the description of the two systems will be very different and the subsequent neural network will have to learn to account for this underlying symmetry.
>
> In the end, we care that our inputs are represented consistently (invariantly) if their relative differences (translations or rotations) are the same according to the system at hand.
> Whether the rotation transformation to obtain the local coordinate frame is with respect to the intrinsic angular velocity or another invariant quantity (e.g., the angles of acceleration vectors) is secondary.
>
> Thus, applying both translation and rotation transformation helps because -from a data-centric point of view- roto-translated local coordinate frames will now allow for coherent regression of the future positions and velocities.
> The reason is that all predictions will be made relative to the present directions of the velocity vectors.
>
> To cast away, however, any remaining doubts, as suggested, we include a new ablation experiment on the charged particles system.
> In this experiment we only use the translation transformation to form local coordinate frames.
> Results are in the table below and corroborate the intuition that rotation transformations are useful also for the point particles.
>
> Method|MSE@t=1|MSE@t=10|MSE@t=20|
> -----:|-------|--------|--------|
> LoCS (Ours)             |3.54e-03±9.93e-05|5.34e-02±5.63e-03|1.22e-01±1.31e-02|
> LoCS (Translation Only) |5.22e-03±2.13e-04|9.32e-02±3.19e-03|2.22e-01±5.40e-03|
> dNRI                    |6.04e-03±1.89e-04|9.25e-02±6.00e-03|4.38e-01±2.89e-01|
> NRI                     |6.02e-03±1.31e-04|2.49e-01±2.45e-02|6.50e-01±4.77e-02|
> EGNN                    |3.52e-03±3.55e-04|1.30e-01±1.26e-02|3.08e-01±2.29e-02|
>
>
> **The writing of the paper need to be improved to make it easier for readers to appreciate it.**
>
> We apologize for the unclarities.
> We will incorporate all your suggestions.
> Also, for Figure 1 we will orientate the cartoons accordingly.
> We will also ask colleagues to do more critical passes of the submission to improve presentation.
>
> References
> ---
>
> Kipf, Thomas, Ethan Fetaya, Kuan-Chieh Wang, Max Welling, and Richard Zemel. "Neural relational inference for interacting systems." In International Conference on Machine Learning, pp. 2688-2697. PMLR, 2018.
>
> Bock, Julian, Robert Krajewski, Tobias Moers, Steffen Runde, Lennart Vater, and Lutz Eckstein. "The ind dataset: A drone dataset of naturalistic road user trajectories at german intersections." In 2020 IEEE Intelligent Vehicles Symposium (IV), pp. 1929-1934. IEEE, 2020.

---

> > ### Comment · Reviewer_49Zp · 2021-08-19
> > **keep my score unchanged.**
> >
> > Thank the authors for addressing my concern. The 2D simple case did give me some intuitions on why roto-translated local coordinate frames allows efficient learning even for particles. Still I'd like to say that for the two given systems, whether roto-translation canonicalization would result in similar representation depends on the direction of velocity of A. If, for example, the velocities of A are in opposite directions in the two systems, the canonicalized systems would have the same representation (at least the position representation would be the same, the velocity representation of course also depends on the velocity of B and C). For other velocity configurations of A, the existence of rotation transformation might not be as important. From the additional experiment results, the rotation transformation did help. This should definitely be added to the revised manuscript.

---

> > > ### Author Response · Authors · 2021-09-01
> > > **Author Response to Reviewer 49Zp**
> > >
> > > We thank the reviewer for the encouraging words and positive feedback. Please find our answers below.
> > >
> > > You are right in that the canonicalized systems would have the same (position) representation when the velocities of A are in opposite directions.
> > > In general, though,
> > > different velocity configurations across different examples may result in different dynamics, and our method will output different local coordinate frames.
> > > That is, changing only a single object's velocity results in a new dynamical system; there is no linear transformation that can transform one system to the other.
> > >
> > > For example, consider the 2 different toy dynamical systems shown below, where arrows represent velocities.
> > > They only differ in particle A's velocity direction.
> > > However, their dynamics vary greatly: in example \#2a, particles A and B are moving perpendicularly to one another and may crush due to attractive forces; that is not the case for the same particle pair in example \#1.
> > >
> > > In the roto-translated local coordinate frame in \#2b, the target particle A is located at the origin and its velocity vector matches the direction of the x-axis.
> > > This is the case for target particles in all local coordinate frames and removes one degree of freedom from modelling interactions, thus making learning more efficient.
> > > This also makes trajectory forecasting easier, since predictions on the x-axis mean that particles continue on a straight trajectory.
> > > Moreover, in example \#2b, the fact that particle B is in the upper right quadrant moving downwards, in and of itself instantly hints at meaningful interactions.
> > > On the other hand, a model operating on translated-only local coordinate frames would have a harder task learning such interactions.
> > > Since representations across local coordinate frames share a common meaning across the dataset, roto-translated coordinate frames result in more efficient learning.
> > >
> > > Example \#1, A's translated-only local coordinate frame
> > > ~~~~
> > > C
> > > ↓  A→ (0,0)
> > >
> > >
> > >     ←B
> > > ~~~~
> > >
> > > Example \#2a, A's translated-only local coordinate frame
> > > ~~~~
> > >                    y
> > > C                  ↑
> > > ↓  A (0,0)         |→ x
> > >    ↓
> > >
> > >     ←B
> > > ~~~~
> > >
> > > Example \#2b, A's roto-translated local coordinate frame
> > > ~~~~
> > >
> > >         B
> > >         ↓          y
> > >                    ↑
> > >    A→ (0,0)        |→ x
> > > C→
> > > ~~~~

---

> > > > ### Comment · Reviewer_49Zp · 2021-09-01
> > > > **increase score to 7**
> > > >
> > > > This is a nice toy example that explains the intuition and motivation behind roto-translation and why it performs better. I encourage the authors to incorporate such explanation into the paper to offer readers more intuition behind mathematical formulations.

---

> > > > > ### Author Response · Authors · 2021-09-02
> > > > > **Thank you**
> > > > >
> > > > > Thank you very much for your kind words and your effort, as well as for raising your score, it is much appreciated!

---

### Official Review · Reviewer_QAhm · 2021-07-16

**Rating:** 7
**Confidence:** 3

**Summary:**

This paper proposes a to use local coordinate system per object in a spatio-temporal graph (with a GNN) of multiple object in order to learn and predict the dynamics of the objects interaction. This enables adding invariance, equivariance, and use anisotropic filters between GNN layers to improve learning performance.


**Limitations And Societal Impact:**

Limitations were adequately discussed.

**Main Review:**

Strengths
+ The addition seems simple and useful for a wide range of problems, at least ones that use graph node features as euclidean states of position and velocity.
+ Experiments demonstration notable improvement over baselines.
+ Paper is mostly well motivated (exploit invariance and symmetry for dynamic tasks, prior work is focused on static datasets) and clearly explained.

Weaknesses
- The overall contribution falls a bit on the weaker side. Besides the insight to swap to a local coordinate system, Section 3.1 is mostly simple linear algebra and rigid body mechanics, commonly employed in graphics, vision, and robotics. Related to that, since rotation matrices are already used why not consider working with SE2 / SE3 Lie groups and make the math less clunky?

- Section 3.2, 'In practice, we do not always have perfect information about the object states,' what application is this with respect to? Is the idea to find the next state from the current velocity i.e. the temporal model assumes a linear dynamical system?

- An example in the introduction would help ground where symmetry arises and why it leads to subpar leaning. '... exhibit symmetries that if left to their own devices lead models to subpar learning.'

- More insight with examples would be helpful in explaining how the paper converged to idea of speed normalization?


**Time Spent Reviewing:**

3

---

> ### Author Response · Authors · 2021-08-10
> **Author Response to Reviewer QAhm**
>
> We thank the reviewer for the encouraging words and positive feedback. Please find our answers below.
>
> **'In practice, we do not always have perfect information about the object states,' what application is this with respect to? Is the idea to find the next state from the current velocity i.e. the temporal model assumes a linear dynamical system?**
>
> By 'perfect information about the object state' we mean to say that we do not always have available the angular positions of the objects.
> This can be either because angular positions are simply not recorded in the datasets available to us, or, because the objects in the experimental setting are seen as point masses (e.g., in the charged particle dataset [Kipf et al., 2018]), in which case an angular position does not make sense to begin with.
>
> When this information is not available, we propose to approximate angular positions using the angles of the velocity vectors.
> The angular positions at each time-step are computed as a function of the velocities of the same time-step, which one could say assumes a linear dynamical system.
> Replacing the unavailable angular positions with angles of velocity vectors works just as well because -in the end- we mainly care to have a consistent (invariant) direction with which to define the local coordinate frame across time steps.
>
> **More insight with examples would be helpful in explaining how the paper converged to idea of speed normalization?**
>
> Standard normalization techniques, such as min-max normalization, perform a translation transformation followed by scaling on the input vector.
> This would apply to velocities as well, which we think is counter-intuitive since they are treated as part of the same feature vector and not as geometric entities.
>
> For one, the translation operations in the min-max normalization on the velocities remove any notion of speed from the input to the neural network, thus making it hard to model future time steps.
> This is because translation in this case equals a vector subtraction, an operation that changes both the magnitude and the direction of vectors.
> What is more, the scaling operations apply anisotropic transformations, affecting each axis differently.
> In the end, from a geometric point of view both operations can change the directions of the velocity vectors and the estimated orientations in an inconsistent manner between examples, and make learning noisy and harder.
>
> Instead, we opt for a data normalization operation that is more geometrically oriented and better in line with local coordinate frames.
> Local coordinate frames naturally tend to center data around the origin; besides, local coordinate frames are invariant to a mere isotropic translation normalization to the node positions.
> For the scaling operation, we opt for a simple isotropic transformation that shrinks relative positions and velocities equivalently across all axes.
>
> **'... why not consider working with SE2 / SE3 Lie groups and make the math less clunky?'**
>
> In this work we sided with simplicity since the results in experiments were already positive enough.
> Incorporating SE2/SE3 is an excellent suggestion though, that we will work on next.
> In the final version of the paper we will improve presentation and reduce clunkiness.
> Thank you for the suggestion!
>
> **'An example in the introduction would help ground where symmetry arises and why it leads to subpar leaning. '... exhibit symmetries that if left to their own devices lead models to subpar learning.''**
>
> Indeed, an example would clarify the narrative.
> We will add an example of traffic trajectories and charged particles to motivate where symmetries might arise and may lead to subpar learning.
>
> References
> ---
>
> Kipf, Thomas, Ethan Fetaya, Kuan-Chieh Wang, Max Welling, and Richard Zemel. "Neural relational inference for interacting systems." In International Conference on Machine Learning, pp. 2688-2697. PMLR, 2018.
>
> Bock, Julian, Robert Krajewski, Tobias Moers, Steffen Runde, Lennart Vater, and Lutz Eckstein. "The ind dataset: A drone dataset of naturalistic road user trajectories at german intersections." In 2020 IEEE Intelligent Vehicles Symposium (IV), pp. 1929-1934. IEEE, 2020.

---

> > ### Comment · Reviewer_QAhm · 2021-09-03
> > **response to authors**
> >
> > Thank you for the detailed response. Having read the reviews and rebuttal, I have increased my score to accept.

---

> > > ### Author Response · Authors · 2021-09-03
> > > **Thank you**
> > >
> > > Thank you very much for your kind words and your effort, as well as for raising your score, it is much appreciated!

---

### Official Review · Reviewer_Yjm7 · 2021-07-16

**Rating:** 7
**Confidence:** 4

**Summary:**

This paper proposes the use of local coordinate frames for dynamics modelling in graph neural networks. 6dof pose, with angles estimated using translational velocity, is used to determine object pose, and graphs constructed in coordinate frames of these objects. Standard GNN dynamics models can then be trained, with the addition of an anisotrophic filter. Results on a range of experimental settings (synthetic, charged particles, mocap) show that dynamics modelling in local coordinate frames is highly effective and results in improved performance when compared with the arbitrary global frames currently in use.

**Ethical Concerns:**

I have no ethical concerns with this work.

**Limitations And Societal Impact:**

The paper briefly discussed limitations, although I would have liked more detail on failure modes, downsides - when will it fail, is it vulnerable to gimbal lock, is the pose estimation affected by symmetry's, are there better orientation representations, etc.

There may be potential societal impacts for tracking applications which could adopt this work, but I believe this work is still in such an early research stage that this is unlikely to be the case.

**Main Review:**

Originality:

The idea of dynamics modelling in local coordinate frames is not new, but this is the first work I have seen to do so in a graph neural network.

Quality:

The paper is clear and well structured, and experiments are concise but thorough, clearly highlighting the value of the proposed approach. Ablations show that coordinate frame changes bring about the greatest improvement in performance. This was the best paper in my set.

Clarity:

The paper is clearly laid out, but I found it extremely formally written, and in many cases simple ideas are described using jargon and terminology that "...obfuscates and impresses rather than clarifies."  to quote troubling trends in machine learning scholarship, Lipton 18. This is an extremely simple, but clearly effective idea, but I found that the paper hides this under layers of unnecessary complexity.

Significance:

I believe the performance improvements obtained by using local coordinate frames are significant enough to warrant publication, and that there is enough in this paper for interested parties to build on.

**Time Spent Reviewing:**

2

---

> ### Author Response · Authors · 2021-08-10
> **Author Response to Reviewer Yjm7**
>
> We thank the reviewer for the encouraging words and positive feedback. Please find our answers below.
>
> **More detail on failure modes, downsides - when will it fail, is it vulnerable to gimbal lock, is the pose estimation affected by symmetry's, are there better orientation representations, etc.**
>
> We have identified two modes where local coordinate frames do not exhibit the same large improvements.
> The first one is when the data setting relies, in fact, on a global coordinate system.
> A characteristic example is the basketball dataset [Yue et al., 2014].
> In the basketball court different coordinates carry different meaning.
> For instance, around the 3-pointer corners the behaviors of the players are different from when they are around the basket.
> While local coordinate frames do not hurt performance when global coordinates have meaning, they do not necessarily improve it either.
> A simple combination of global and local coordinate systems worked well in our experiments, however, we left that experiment out for the sake of clarity.
>
> The second mode is when the direction of the local coordinate frames cannot be well defined, as reviewer fJBY correctly pointed out.
> For instance, when the velocity of an object is 0, there is -in theory- no velocity vector to define the local frame.
> In practice, we simply (and arbitrarily) set the orientation to 0.
> This worked just as well, also because objects with 0 velocity tended to carry low inductive information, at least in the datasets available to us.
>
> We will explain more clearly these two limitation modes in the final version of the paper.
>
> Euler angles are indeed prone to singularities and gimbal lock.
> As with other methods relying on similar geometric rotations, local coordinate frames could somehow be adversely affected as well.
> In practice, we never experienced any issues with training or evaluating the method in any of our experiments, nor did we observe predicted trajectories that would point to singularities or gimbal lock.
> That is probably because in the end, we only use Euler angles as part of the feature vectors, which are then fed to the graph neural network; we did not use the angles as a representation themselves.
> That said, it would be very interesting to investigate different orientation representations that do not suffer from singularities, such as quaternions.
>
> **The paper is clearly laid out, but I found it extremely formally written.**
>
> We agree.
> Our goal is definitely not to obfuscate, rather clearly lay out the form local coordinate frames take such that it is close to the actual implementation.
> In the final version, we will include an algorithm box that explains the process in simple steps.
> Thank you!
>
> References
> ---
>
> Yue, Yisong, Patrick Lucey, Peter Carr, Alina Bialkowski, and Iain Matthews. "Learning fine-grained spatial models for dynamic sports play prediction." In 2014 IEEE international conference on data mining, pp. 670-679. IEEE, 2014.

---

> > ### Comment · Reviewer_Yjm7 · 2021-09-02
> > **Thanks for this**
> >
> > By the way, thanks for the detailed response, I still think this is a good paper and my original evaluation stands.

---

> > > ### Author Response · Authors · 2021-09-02
> > > **Thank you**
> > >
> > > Thank you very much for your kind words and your effort.

---

### Decision · Program_Chairs · 2021-09-27

**Decision:**

Accept (Poster)

**Comment:**

As all reviewers agree, the paper discusses an interesting approach to the problem of learning complex dynamical systems with notable experimental supports. Although some reviewers have concerns on the technical part, the authors' responses have resolved those. The paper still have some weak points, especially in the presentation, but I think I can expect the authors modify the paper in the camera-ready by reflecting the discussion. Based on these, I recommend acceptance (poster) for this paper.